# Flamingo participates in multiple models of cell competition

Pablo Sanchez Bosch[†], Bomsoo Cho[†], Jeffrey D Axelrod*

Department of Pathology, Stanford University School of Medicine, Stanford, United States

## eLife Assessment

This study investigates the role of the Cadherin Flamingo (Fmi) in cell competition in developing tissues in *Drosophila melanogaster*. The findings are **valuable** in that they show that Fmi is required in winning cells in several competitive contexts. The evidence supporting the conclusions is **solid**, as the authors identify Fmi as a potential new regulator of cell competition, however, they don't delve into a mechanistic understanding of how this occurs.

**\*For correspondence:**
jaxelrod@stanford.edu

[†]These authors contributed equally to this work

**Competing interest:** The authors declare that no competing interests exist.

## Abstract

The growth and survival of cells with different fitness, such as those with a proliferative advantage or a deleterious mutation, is controlled through cell competition. During development, cell competition enables healthy cells to eliminate less fit cells that could jeopardize tissue integrity, and facilitates the elimination of pre-malignant cells by healthy cells as a surveillance mechanism to prevent oncogenesis. Malignant cells also benefit from cell competition to promote their expansion. Despite its ubiquitous presence, the mechanisms governing cell competition, particularly those common to developmental competition and tumorigenesis, are poorly understood. Here, we show that in *Drosophila*, the planar cell polarity (PCP) protein Flamingo (Fmi) is required by winners to maintain their status during cell competition in malignant tumors to overtake healthy tissue, in early pre-malignant cells when they overproliferate among wildtype cells, in healthy cells when they later eliminate pre-malignant cells, and by supercompetitors as they compete to occupy excessive territory within wildtype tissues. 'Would-be' winners that lack Fmi are unable to overproliferate, and instead become losers. We demonstrate that the role of Fmi in cell competition is independent of PCP, and that it uses a distinct mechanism that may more closely resemble one used in other less well-defined functions of Fmi.

## Introduction

Dividing cells in proliferative tissues must maintain tissue homeostasis and structural integrity as well as preserve their genetic integrity. One of the mechanisms operating in proliferative tissues to achieve these requirements is cell competition. Cell competition is a process in which cells of a higher fitness ('winners') eliminate less fit neighbors ('losers'), by inducing cell death (***Norman et al., 2012***; ***Meyer et al., 2014***; ***Di Giacomo et al., 2017***; ***Watanabe et al., 2018***) and/or extrusion from the epithelial layer (***Norman et al., 2012***; ***Kon et al., 2017***; ***Kohashi et al., 2021***). Cell competition was first recognized in 1975 (***Morata and Ripoll, 1975***), when Morata and Ripoll studied the growth of *Drosophila* carrying a mutation in the gene for the ribosomal protein RpS17 (termed *minute*). Animals heterozygous for this *minute* mutation are viable but develop slowly due to decreased proliferation rates. They further observed that clones of cells carrying this *minute* mutation in a wildtype background were eliminated from the *Drosophila* developing wing. Since then, extensive advances have been made in our understanding of the process (reviewed in ***Morata, 2021***). We now know that cell competition can

be triggered by mutations that cause a disadvantage, such as the aforementioned *minute* cells, and can also be triggered by mutations that confer a proliferative advantage, such as ectopic expression of proto-oncogenes such as Myc or Ras (*Moreno and Basler, 2004*; *de la Cova et al., 2004*; *Hogan et al., 2009*). In this case, the process is known as supercompetition, and wildtype cells behave as losers and are eliminated from the tissue by the mutant supercompetitors. The 'survival of the fittest' effect observed in cell competition has been shown to be critical during embryonic and larval development as well as for tissue homeostasis (*Amoyel and Bach, 2014*; *Maruyama and Fujita, 2017*; *Morata, 2021*). Much additional research has shown that cells and their neighbors are constantly evaluating their fitness, and that the active elimination of less fit cells is critical to maintain tissue integrity (*Ellis et al., 2019*), maintain chromosomal stability (*Baillon et al., 2018*; *Ji et al., 2021*), ameliorate effects of cellular aging (*Merino et al., 2015*), and suppress tumorigenesis (*Kon et al., 2017*; *de Vreede et al., 2022*).

While cell competition has been observed in many cell types, much attention has focused on studies in epithelia (*Vincent et al., 2013*). Polarity is a hallmark of epithelial tissues, and loss of polarity plays a prominent role in triggering cell competition and in the process of eliminating loser cells. Cells harboring mutations in apicobasal polarity genes have been shown to induce and/or regulate cell competition when surrounded by wildtype cells. In *Drosophila*, mutations in apicobasal polarity genes *scribble* (*scrib*), *discs large* (*dlg*), and *lethal giant larva* (*lgl*) all trigger cell competition when induced clonally (i.e. surrounded by wildtype cells), and the clones are eliminated from the epithelial tissue (*Hariharan and Bilder, 2006*; *Morata and Calleja, 2020*). However, clones that are deficient for any of these polarity genes can survive if they are competing against less fit cells, or if they carry additional mutations that confer a growth advantage. For example, when *lgl* clones, which would be eliminated when surrounded by wildtype cells, are made to express elevated levels of Myc, the competition is reversed, and they instead become winners (*Froldi et al., 2010*). This context dependence and fitness surveillance highlights the complexity of cell competition and its plasticity.

In epithelial tissues, cell competition is critical to suppress tumorigenesis, as polarity-deficient clones are prone to malignancy. When *scrib* mutant epithelial cells acquire elevated activity of a proto-oncogene such as the constitutively active *Ras^V12^*, these clones acquire malignant properties of ectopic growth and invasiveness (*Pagliarini and Xu, 2003*; *Brumby and Richardson, 2003*). Surveillance and rapid elimination of cells that develop polarity defects therefore serves to protect epithelia from oncogenesis. This competitive mechanism is known as epithelial defense against cancer (EDAC) (*Maruyama and Fujita, 2017*; *Kon and Fujita, 2021*). Through EDAC, normal epithelial cells eliminate neighboring transformed cells by inducing apoptosis or by inducing their apical or basal extrusion from the epithelial layer (*Watanabe et al., 2018*; *Kohashi et al., 2021*). On the other hand, tumors that exhibit more aggressive properties and escape the defense mechanisms not only proliferate but actively engage in cell competition to promote their own growth and malignancy (*Vishwakarma and Piddini, 2020*; *Madan et al., 2022*). For example, growth of *Drosophila* APC^-/-^ intestinal adenomas requires active elimination of wildtype cells, and inhibiting cell death in the wildtype tissue hampers adenoma growth (*Suijkerbuijk et al., 2016*).

In addition to apicobasal polarity, epithelial tissues are planar polarized. Planar cell polarity (PCP) signaling polarizes cells within the plane of the epithelium to orient cellular structures, cell divisions, and cell migration during development and homeostasis (*Adler, 2002*; *Simons and Mlodzik, 2008*; *Vladar et al., 2009*; *Butler and Wallingford, 2017*). Intercellular Fmi homodimers scaffold the assembly of a set of core PCP proteins: on one side of a cell, Frizzled (Fz), Dishevelled (Dsh), and Diego (Dgo) comprise a complex that interacts with another complex containing Van Gogh (Vang) and Prickle (Pk) on the opposite side of the adjacent cell. Fmi homodimers transmit differential signals in opposite directions to communicate the presence of either protein complex to the other, linking the proximal and distal complexes and mediating asymmetric signaling between them (*Lawrence et al., 2004*; *Strutt and Strutt, 2007*; *Strutt and Strutt, 2008*; *Chen et al., 2008*; *Struhl et al., 2012*). In addition to establishing planar polarity, numerous reports have suggested links between PCP signaling and cancer progression, promoting cell motility, invasiveness, and metastasis (*Weeraratna et al., 2002*; *Katoh, 2005*; *Katoh and Katoh, 2007*; *Coyle et al., 2008*; *Gujral et al., 2014*; *VanderVorst et al., 2018*; *VanderVorst et al., 2023*). Its roles, however, remain poorly characterized.

To probe a potential connection between PCP and cancer, we employed a *Drosophila* eye tumor model. We discovered a requirement for Fmi, but not other core PCP proteins, in tumor-associated

cell competition. We found that aggressive tumors require Fmi to outcompete the neighboring wild-type tissue. We then found that this Fmi requirement is not unique to tumor competition; in several developmental cell competition scenarios, removing Fmi from winner cells prevents them from eliminating their neighbors and transforms them into losers. 'Would-be' winners that lack Fmi show both a decrease in proliferation and an increase in apoptosis. By several criteria, we show that this function for Fmi is independent of its role in PCP signaling.

## Results

### Flamingo is required for tumorigenesis in *Drosophila* Ras$^{V12}$ tumors

Planar polarity has been linked to tumorigenesis in several organisms, tumor models, and patient tumor samples (*Kauck et al., 2013*; *Asad et al., 2014*; *Puvirajesinghe et al., 2016*; *VanderVorst et al., 2018*; *Chen et al., 2021b*; *Li et al., 2021*; *Chen et al., 2021a*). We explored how the core PCP complex might be involved in tumorigenesis. To do so, we used the versatile and widely used Ras$^{V12}$ tumor model in *Drosophila* eye imaginal discs. In this model, eye disc cells are transformed into highly tumorigenic cells by co-expressing the constitutively active Ras$^{V12}$ oncoprotein and RNAi against the tumor suppressor Scribble (Scrib) (*Pagliarini and Xu, 2003*). Cells expressing these genes quickly expand into massive tumors and invade the neighboring brain tissue (*Figure 1A–C*).

To study the requirement for PCP signaling in tumors, we expressed *Ras$^{V12}$* and *scrib* RNAi throughout the eye, and simultaneously downregulated PCP core proteins by also expressing validated RNAis against *fmi*, *fz*, *dsh*, *vang*, and *pk*. We then imaged tumor size through the pupal cuticle. We observed that tumor growth was unaffected by knockdown of any of the core PCP genes at third instar larval or at early pupal stages (*Figure 1A–H*). We then asked whether PCP signaling might be required by tumors when they're confronted by wildtype cells. To test this hypothesis, we created clonal tumors in the eye disc by expressing *Ras$^{V12}$* and *scrib* RNAi under control of ey-Flp. In this model, nearly every cell in the eye disc will perform chromosomal recombination (*Figure 1—figure supplement 1*), producing an eye disc composed of approximately 50% wildtype and 50% tumor cells. Under these conditions, tumor cells expand aggressively, displacing wildtype cells and causing massive overgrowth of the eye disc and metastasis to the larval brain (*Figure 1I–K*). We created clonal tumors that were also null for either *fmi*, *fz*, *dgo*, *vang*, or *pk* and evaluated them for their ability to grow and metastasize when confronted with wildtype cells (*Figure 1I–P*). Remarkably, only *fmi* interfered with tumorigenesis, restraining growth of tumors in the eye disc and preventing them from invading neighboring tissues (*Figure 1L*). To ensure that the growth inhibitory effect of loss of Fmi in clonal tumors but not in whole eye tumors was not a result from comparing a null allele in clonal tumors vs RNAi in whole eye tumors, we generated *fmi* RNAi clonal tumors and compared them to control RNAi clonal tumors. While the effect was not as dramatic as with null clones, the size of *fmi* RNAi clonal tumors was substantially reduced compared to that of clonal tumors expressing *w* RNAi (*Figure 1—figure supplement 1A and B*). Therefore, the same partial level of fmi knockdown impairs clonal tumor growth but not growth of whole eye tumors.

We also wished to directly compare growth of *fmi*-null clonal and *fmi*-null whole eye tumors. Because *fmi* alleles are embryonic lethal, we generated *fmi*-null tumors while eliminating WT cells from the eye disc using the GMR-Hid, l(2)CL system (*Stowers and Schwarz, 1999*). While *fmi*-null clonal tumors showed restrained growth (*Figure 1L*), elimination of WT cells in eye discs removed competition, and *fmi*-null tumors grew to a size comparable to control whole eye tumors (*Figure 1—figure supplement 1C and D*). Because the effect of *fmi* was only apparent when tumors were induced in juxtaposition to wildtype cells, we hypothesized that Fmi may only be required in tumors that undergo cell competition.

### Flamingo is required in winner cells during cell competition

Molecular mechanisms used by tumors in cell competition are sometimes shared by developmental cell competition. Malignant cells have been shown to actively eliminate surrounding cells by mechanical cell competition, apoptosis, and engulfment (*Levayer et al., 2016*; *Kohashi et al., 2021*), similar to the cell death-mediated elimination of losers during developmental cell competition and EDAC (*Maruyama and Fujita, 2017*; *Parker et al., 2021*) Induction of JNK-mediated cell death is also common to developmental competition and tumor cell competition, as shown in intestinal adenomas

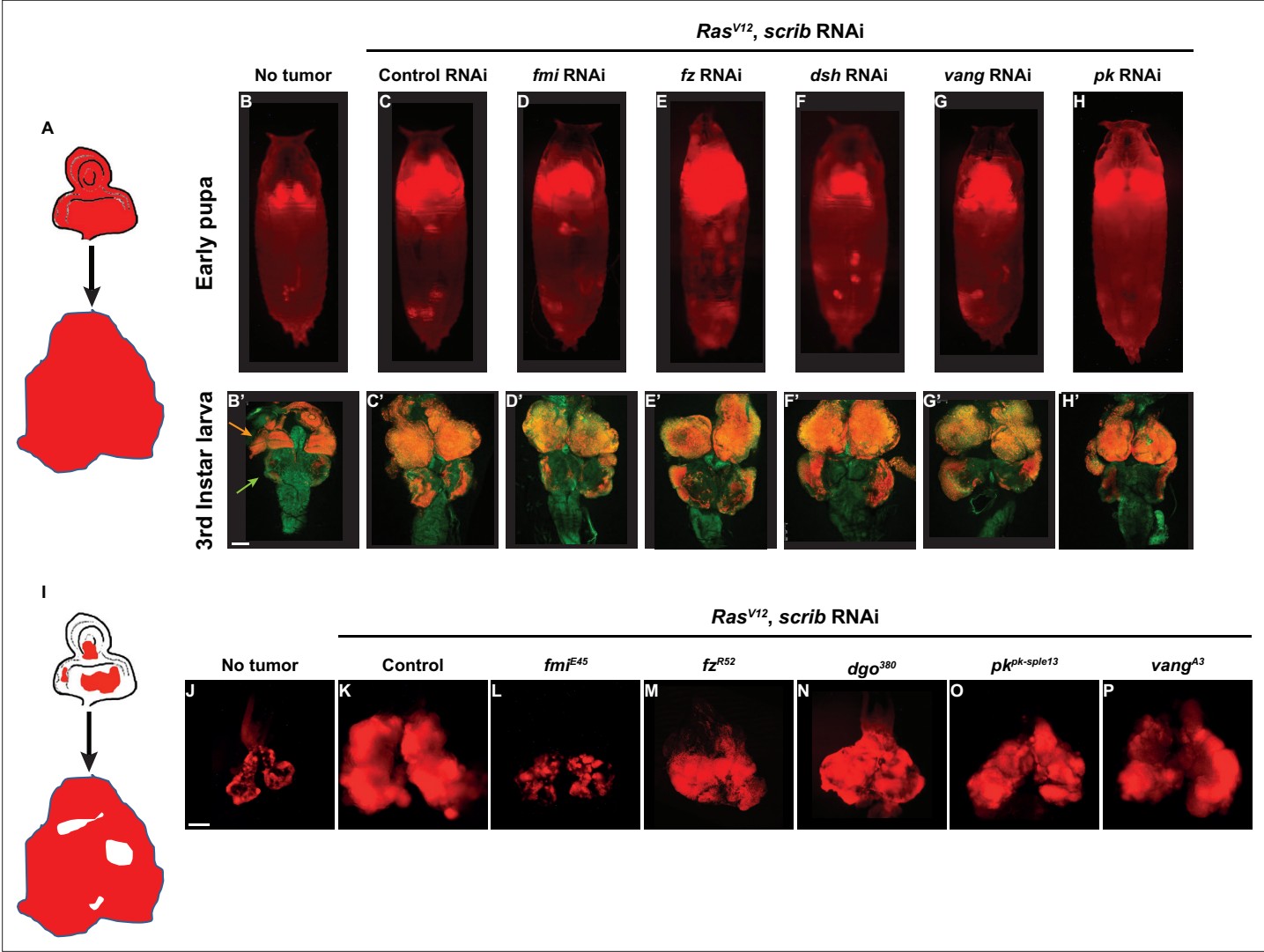

**Figure 1.** Fmi is required in clonal tumors to outcompete wildtype tissue. (**A**) Schematic of the whole eye disc Ras$^{V12}$, scrib RNAi, RFP tumors. (**B–H**) Pupal RFP tumors imaged through the cuticle. All pupae are ey-Flp; act5C>CD2>Gal4, UAS-RFP with the indicated RNAi. N=5 (no tumor), 10 (control RNAi), 13 (fmi RNAi), 17 (fz RNAi), 19 (dsh RNAi), 17 (vang RNAi), and 8 (pk RNAi). (**C**) Control UAS-Ras$^{V12}$, UAS-scrib RNAi, UAS-w RNAi tumors. (**D–H**) UAS-RasV12, UAS-scrib RNAi tumors co-expressing UAS-RNAi against the planar cell polarity (PCP) genes indicated above each pupa. (**B'–H'**) Representative third instar larval brain and eye discs for each of the experimental groups from above. Arrowheads in B' point to the eye-antenna imaginal disc (orange) and the brain lobes (green). N=3 (no tumor), 10 (control RNAi), 13 (fmi RNAi), 12 (fz RNAi), 15 (dsh RNAi), 20 (vang RNAi), and 10 (pk RNAi). (**I**) Schematic of eye disc Ras$^{V12}$, scrib RNAi, RFP clonal tumors. (**J–P**) Third instar larval eye discs RFP tumor clones generated via ey-Flp; FRT42D Gal80/FRT42D; act5C>CD2>Gal4, UAS-RFP. (**J**) Eye discs with non-tumor, control RFP clones. (**K**) Control Ras$^{V12}$, UAS-scrib RNAi, RFP clonal tumors. (**L–P**) Ras$^{V12}$, scrib RNAi, RFP clonal tumors carrying the PCP allele indicated above each panel. Scale bars: 100 μm.

The online version of this article includes the following figure supplement(s) for figure 1:

**Figure supplement 1.** The ey-Flp recombination system causes almost every cell in the eye disc to undergo chromosomal recombination.

**Figure supplement 2.** fmi RNAi co-expression decreases the size of clonal Ras$^{V12}$, scrib RNAi tumors.

---

(*Suijkerbuijk et al., 2016*). We hypothesized that the requirement of Fmi by competing tumors might also be shared in developmental cell competition. We therefore evaluated the role of Fmi in non-malignant, developmental cell competition models.

Cells expressing high levels of Myc behave as winners when growing among wildtype cells, proliferating faster than wildtype cells and inducing their elimination. Despite Myc being a proto-oncogene, in contrast to Ras$^{V12}$ scrib RNAi tumors, Myc overexpressing clones do not become tumors, but instead expand to occupy larger than normal domains of morphologically normal tissue (*Moreno and Basler, 2004*; *de la Cova et al., 2004*; *Clavería et al., 2013*). As for the Ras$^{V12}$ scrib RNAi tumor

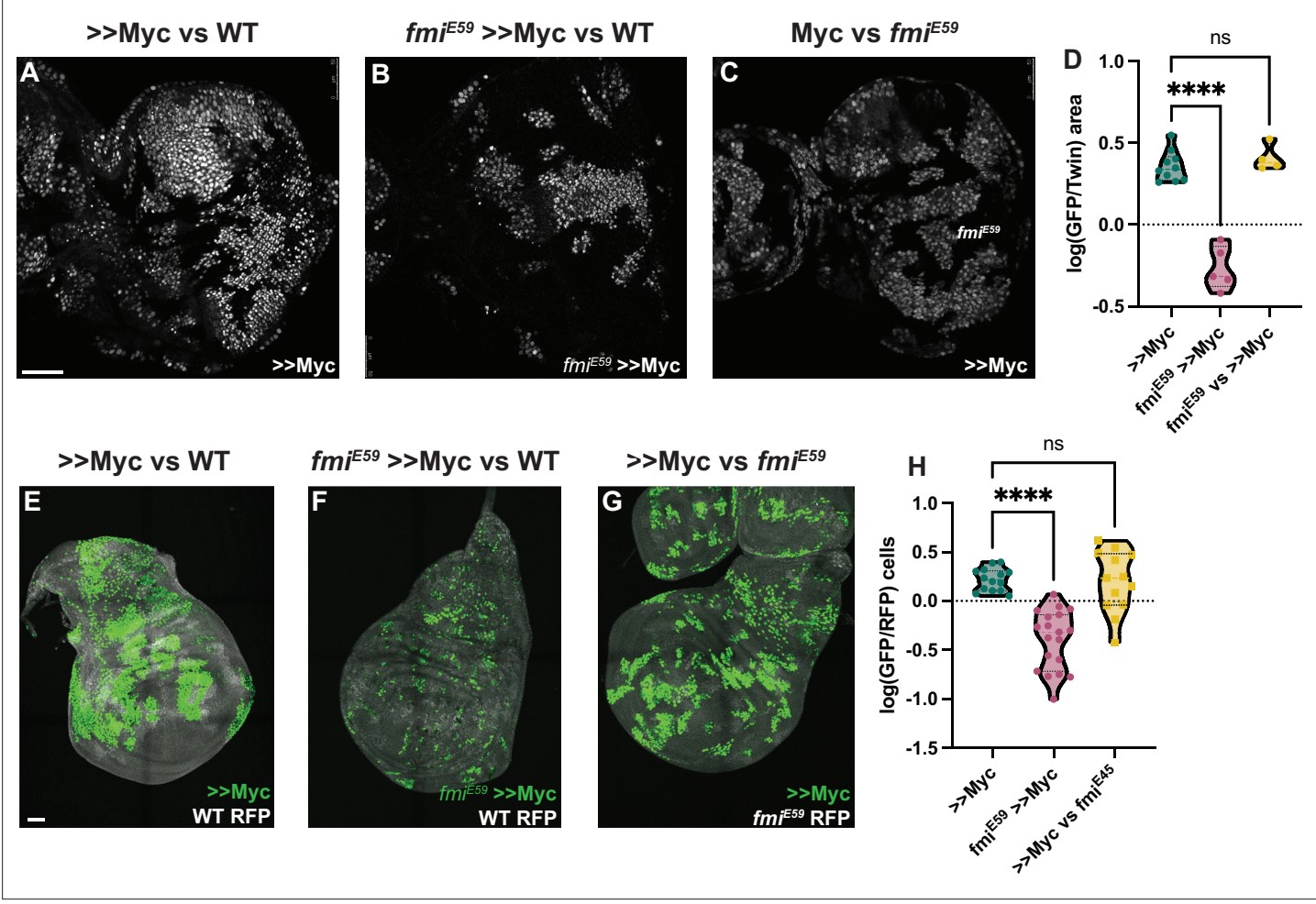

**Figure 2.** Winners require Fmi in developmental supercompetition. (**A**) Eye imaginal disc clones overexpressing UAS-Myc (>>Myc) and UAS-nGFP under control of the tub >CD2>GAL4 driver. Clones were generated with ey-Flp, so half of the cells are clones and the other half are wildtype twins. (**B**) Eye imaginal disc >>Myc, UAS-nGFP, *fmi*$^{E59}$ clones, competing against wildtype twins. (**C**) Eye imaginal disc >>Myc, UAS-nGFP clones, competing against *fmi*$^{E59}$ twins. (**D**) Ratio of RFP-labeled clone area vs unlabeled wildtype area. The unlabeled wildtype area was obtained by subtracting the RFP-labeled area from the total eye disc area. A ratio over 1 implies RFP-positive clones are over-represented, likely behaving as supercompetitors, while a ratio below 1 means the RFP cells are under-represented, likely losers. N=9 discs (>>Myc), 5 discs (*fmi*$^{E59}$,>>Myc), 4 discs (*fmi*$^{E59}$ vs >>Myc), groups were analyzed using multiple unpaired, two-tailed t-test; p-values:<0.0001 (****), 0.6340 (ns). (**E**) Representative wing imaginal disc overexpressing >>Myc and UAS-nGFP under control of the tub-Gal4 driver. Clones were generated using hsp70-Flp, with a 15 min 37°C heat-shock. Wildtype cells are labeled with tub-nRFP. Non-recombinant cells are heterozygous for nRFP, while twin spots are homozygous nRFP. n=14 discs. (**F**) Representative wing imaginal disc with >>Myc, UAS-nGFP, *fmi*$^{E59}$ clones, over a wildtype background. Clones were generated in the same fashion as D. n=19 discs. (**G**) Representative wing imaginal disc with >>Myc, UAS-nGFP clones, competing against *fmi*$^{E59}$ twin spots. n=13 discs. (**H**) log10 of the >>Myc/Twin spot cell ratio. nGFP total cells were divided by the number of twin spot homozygous nRFP cells, and then log10-transformed. A ratio over 0 indicates over-representation (likely supercompetition) of nGFP+ clones, and below 0 indicates nGFP+ cells are under-represented (likely behaving as losers). The difference between groups was analyzed using a one-way ANOVA, with Dunnett correction for multiple comparisons; p-value: >>Myc vs *fmi*$^{E59}$ >>Myc < 0.0001 (****); >>Myc vs *fmi*$^{E59}$=0.9705 (ns). Scale bars: 50 µm.

The online version of this article includes the following figure supplement(s) for figure 2:

**Figure supplement 1.** Fmi by itself does not trigger cell competition.

model, we used UAS-myc and ey-Flp to create eye discs comprising initially equal populations of Myc overexpressing (>>Myc) and wildtype cells. As expected, >>Myc cells behaved like supercompeti-tors, and the majority of cells (~65%) in late third instar discs were >>Myc cells (*Figure 2A*). However, when >>Myc cells were depleted of *fmi*, they failed to outcompete wildtype cells and behaved as one would expect of losers; the >>Myc, *fmi*$^{-/-}$ cells were outcompeted by wildtype cells such that late third instar eye discs were composed of only ~35% >>Myc, *fmi*$^{-/-}$ cells, suggesting that wildtype cells

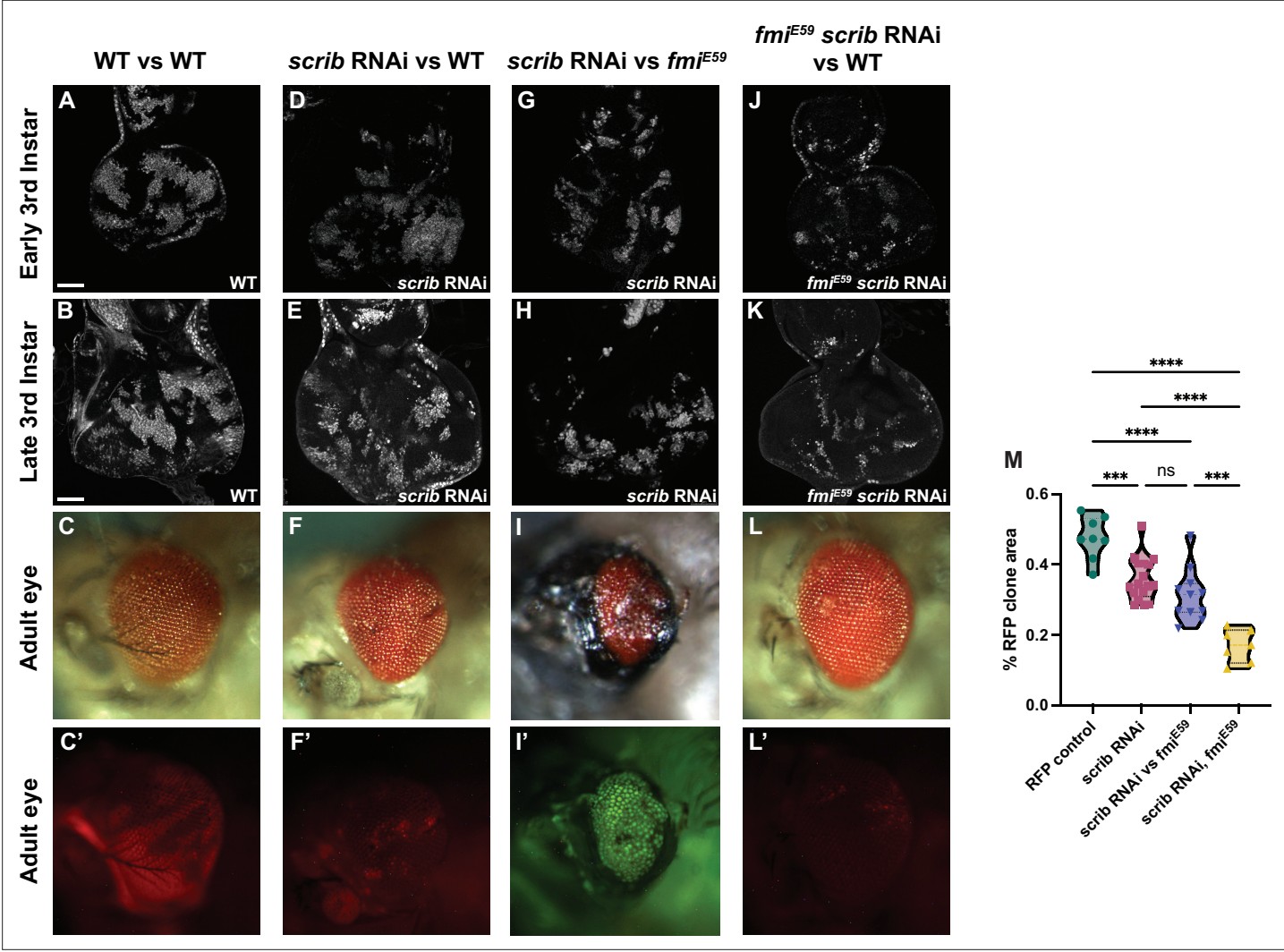

**Figure 3.** Winners require fmi in *scribble* cell competition. (**A–L'**) scrib RNAi clone analysis in the prospective eye. UAS-scrib RNAi was expressed with the act5C>CD2>Gal4 driver and clones were generated by ey-Flp. Clones are marked both with UAS-nGFP and tub-nRFP. (**A**) Representative early third instar disc with control UAS-nGFP clones. (**B**) Representative late third instar disc with control UAS-nGFP clones. (**C, C'**) Representative adult eye with control UAS-nGFP clones. C' shows fluorescently labeled control clone cells. (**D**) Representative early third instar disc with UAS-scrib RNAi clones vs wildtype twin clones. (**E**) Representative late third instar disc with UAS-scrib RNAi clones vs wildtype twin clones. (**F, F'**) Representative adult eye with UAS-scrib RNAi clones vs wildtype twin clones. F' shows fluorescently labeled surviving UAS-scrib RNAi cells. (**G**) Representative early third instar disc with UAS-scrib RNAi clones vs *fmi^E59* twin clones. (**H**) Representative late third instar disc with UAS-scrib RNAi clones vs *fmi^E59* twin clones. (**I, I'**) Representative adult eye with UAS-scrib RNAi clones vs *fmi^E59* twin clones. This phenotype was lethal. The adult eye shown was from an escaper. Escapers had trouble eclosing and died within hours. I' shows fluorescently labeled surviving scrib RNAi cells. tub-nRFP fluorescence was barely visible, so the stronger UAS-nGFP is shown. (**J**) Representative early third instar disc with *fmi^E59*, UAS-scrib RNAi clones vs wildtype twin clones. (**K**) Representative late third instar disc with *fmi^E59*, UAS-scrib RNAi clones vs wildtype twin clones. (**L–L'**) Representative adult eye with *fmi^E59*, UAS-scrib RNAi clones vs wildtype twin clones. L' shows fluorescently labeled surviving *fmi^E59*, UAS-scrib RNAi cells. (**M**) Quantification of fluorescently tagged clone area for the phenotypes mentioned above. Each dot in the violin plot displays the ratio of RFP+ area vs total disc area. Of special significance is the fact that scrib RNAi clones competing with wildtype clones survive similarly to scrib RNAi clones competing with *fmi^E59*. p-Values are as follows: ****<0.0001; ***<0.0005; ns = 0.2634. Scale bars: 50 μm.

The online version of this article includes the following figure supplement(s) for figure 3:

**Figure supplement 1.** Clone size during early third instar *scrib* competition.

behave as winners against >>Myc, *fmi^-/-* cells (*Figure 2B and D*). We then tested whether Fmi had an effect in the losers during >>Myc competition. Removing Fmi from the WT losers had no effect on this competition, as >>Myc clones did not compete more successfully than when confronting wildtype cells (*Figure 3C and D*). Importantly, loss of Fmi alone did not induce competition, as in eye discs

comprising initially equal populations of *fmi*$^{-/-}$ and wildtype cells the *fmi* mutant cells grew equally well as the wildtype cells (*Figure 2—figure supplement 1A–C*).

If the requirement of Fmi is a general feature of cell competition, it should be observable in other tissues. We therefore assessed whether depleting Fmi in wing disc >>Myc winner clones would also reverse the competition outcome. We used hsp-Flp to generate twin spot clones expressing UAS-Myc, and as in eye discs, those clones behaved like supercompetitors. Myc clones were on average 1.7 times larger than the homozygous RFP+ (hRFP) twin spots (*Figure 2E*). However, when >>Myc clones lacked Fmi, their ability to compete was severely impaired, producing much smaller clones, being on average half as large as their hRFP twin spots, and thus, like in eye discs, behaved as one would expect of losers when competing against wildtype cells (*Figure 2F–H*). Interestingly, *fmi*$^{E59}$ Myc clones were also highly fragmented in the wing disc, a phenomenon we did not observe in eye discs. As observed in eye discs, removing Fmi from the wildtype losers had no effect on the ability of >>Myc clones to compete (*Figure 3G and H*). Similarly, making clones mutant for *fmi* alone had no effect on their growth in wing discs (*Figure 2—figure supplement 1D–F*).

In the setting of tumorigenesis, either tumor cells or wildtype cells may emerge as winners of cell competition. Tumors outcompete wildtype cells in the processes of invasion and metastasis, whereas transformed pre-malignant cells are often eliminated by wildtype cells through competition-dependent defense mechanisms such as EDAC (*Kon et al., 2017*; *Kon and Fujita, 2021*). We therefore further explored the potential requirement for Fmi in a system in which both outcomes occur at different times. In eye discs, cells lacking Scrib display a proto-oncogenic behavior, losing polarity and overproliferating in the developing eye disc. However, as eye development progresses, they subsequently become losers, and are eliminated via JNK-mediated apoptosis during late third instar and pupal development so that they are virtually absent from the tissue before the fly ecloses (*Brumby and Richardson, 2003*).

Using the ey-Flp system to activate RNAi against *scrib* in eye discs, we confirmed that downregulation of *scrib* generated clones that perdure through larval development (*Figure 3D , E, and M*) but were eliminated from the tissue by the end of pupal development and were not detected in the adult eye (*Figure 3F and F'*). Compared to control RFP clones, which represented around 40% of the eye cell population in early third instar larva (*Figure 3A, B, and M*, *Figure 3—figure supplement 1A*) and remained at roughly that proportion in the adult eye (*Figure 3C and C'*), *scrib* RNAi clones began to be eliminated at or before the time the morphogenetic furrow progresses (*Figure 3E and M*, *Figure 3—figure supplement 1B*), and were almost completely absent in adult eyes (*Figure 3F and F'*). Their elimination left scars in the adult eye, likely because the structure of the eye is established with the passing of the morphogenetic furrow, and the compound eyes were smaller compared to wildtype eyes (*Figure 3C and F*). Wildtype cells therefore behave as winners when confronted with *scrib* RNAi clones during late larval and pupal development.

We then evaluated the effect of removing Fmi from the wildtype winner clones in *scrib* cell competition. Loss of *fmi* in wildtype cells had little impact on the size of *scrib* RNAi clones in third instar larval discs (*Figure 3G, H, and M*, *Figure 3—figure supplement 1C*), but as development progressed, wildtype winner cells depleted of *fmi* lost their ability to compete with and eliminate *scrib* RNAi clones; the *scrib* RNAi clones survived during pupal development and indeed constituted the majority of the adult eye (*Figure 3I and I'*), showing a reversal in the competition outcome much as the outcome of >>Myc competition is reversed when the winners lack Fmi. The surviving *scrib* RNAi cells failed to differentiate, and instead produced large scars in the adult eye (*Figure 3I and I'*). We also observed considerable lethality in this population of flies, presumably due to the uncontrolled proliferation of *scrib* RNAi cells that could not be eliminated.

During >>Myc supercompetition, removing Fmi from the loser wildtype cells showed no effect in cell competition in eye and wing discs (*Figure 2C and G*). We therefore considered what might happen if *scrib* RNAi loser clones are made to lack Fmi. However, this situation is more complex than the >>Myc competition, since *scrib* RNAi clones have been previously documented to initially overproliferate, perhaps behaving as winners, before ultimately becoming losers (*Brumby and Richardson, 2003*). We were unable to quantify this early proliferation of *scrib* RNAi eye clones before they are eliminated by their wildtype neighbors, as the discs are too small for us to dissect and count. However, assuming that they are behaving as winners in early larval development, one might expect that removing Fmi from *scrib* RNAi clones would impair their ability to overproliferate early, such that

by third instar, they would be smaller than *scrib* RNAi clones with intact Fmi. Indeed, this is what we observed (*Figure 3J, K, and M*, *Figure 3—figure supplement 1*). While some portion of their smaller size relative to control clones is attributed to their elimination by wildtype winners as described above, the remainder may be a result of the loss of their ability to overproliferate earlier in larval development.

The small population of *fmi^E59^ scrib* RNAi clones remaining at third instar was almost completely eliminated by the end of larval development, such that the adult eyes were indistinguishable from wildtype eyes (*Figure 3L and L'*). We hypothesize that the lack of *fmi* in *scrib* clones hinders their ability to overproliferate in early larval stages, resulting in smaller clones that are quickly eliminated by their wildtype neighbors, allowing compensatory differentiation to generate a morphologically normal eye.

## Loss of fmi triggers cell death and reduces proliferation in 'would-be' winner clones

Our observations suggest that lack of *fmi* renders winner clones and tumors incapable of outcompeting the neighboring tissue in multiple cell competition scenarios. Cell competition relies mainly on two mechanisms to allow winner cells to take over the tissue when confronting less fit cells: faster proliferation than losers and elimination of loser cells by induced apoptosis or extrusion (*Amoyel and Bach, 2014*; *Maruyama and Fujita, 2017*; *Morata, 2021*). We therefore explored how the lack of *fmi* in tumors and winner >>Myc cells during competition affected both mechanisms.

*Drosophila* control Ras^V12^, *scrib* RNAi eye tumors trigger apoptosis in the neighboring cells, as detected by Dcp1 staining (*Figure 4A–D*), consistent with previous reports (*Karim and Rubin, 1998*; *Pérez et al., 2017*) and with their highly invasive behavior when surrounded by wildtype cells. However, when *fmi* was removed from the clonal tumors, the outcome of this competition was reversed, as described above, and instead, *fmi^-/-^*, Ras^V12^, *scrib* RNAi tumors displayed excess apoptosis compared to the surrounding wildtype tissue (*Figure 4E–H*). Moreover, we found cell debris from *fmi*-deficient tumor cells inside lysosomal vesicles in wildtype cells, suggesting that wildtype cells are clearing neighboring loser tumor cells through engulfment (*Figure 4—figure supplement 1*). We then tested whether this reversal of apoptosis burden was specific to the tumor model, or if it is a mechanism shared in other cell competition models. We counted apoptotic cells in WT and >>Myc clones in the eye disc only when these cells were located near to the clone boundary (1–3 cells from the boundary, see Materials and methods for details). We observed that control >>Myc clones showed low levels of apoptosis and similar levels in their neighboring WT cells (*Figure 4I–L*; p-value 0.6049), in line with previous results (*de la Cova et al., 2004*), whereas, consistent with the tumor model, apoptosis was significantly increased in *fmi^E59^*, >>Myc clones but not in their WT neighbors (*Figure 4M–P*, p-value 0.0006), likely contributing to the reversal in competition we previously observed (*Figure 2A–D*).

JNK signaling plays a prominent role during cell competition. Previous reports have shown that activation of JNK mediates engulfment and elimination of *scribble* clones during cell competition (*Ohsawa et al., 2011*; *Yamamoto et al., 2017*). However, JNK signaling is also responsible for increased tumor growth and invasion in Ras-activated cells or scrib mutants (*Igaki et al., 2006*; *Uhlirova and Bohmann, 2006*; *Leong et al., 2009*). We therefore asked whether the role of Fmi in tumorigenesis and cell competition could be related to the regulation of JNK signaling. Confirming previous observations, we detect activation of JNK signaling via the *puckered* lacZ reporter *puc^E69^* both in Ras^V12^, *scrib* RNAi tumors (*Figure 4A–C*) and *scrib* RNAi clones in eye discs (*Figure 4—figure supplement 2A–C*). However, Ras^V12^ tumors lacking *fmi*, despite displaying cell death in tumor cells competing with wildtype neighbors, show no change in Puc activation (*Figure 4E–G*). Similarly, *scrib* RNAi clones activate JNK signaling independently of *fmi* (*Figure 4—figure supplement 2D–F*). We therefore conclude that Fmi acts independently of JNK signaling.

Cell competition does not rely solely on apoptosis to eliminate loser cells. An increased proliferation rate of winners is also a hallmark of cell competition (*Morata, 2021*; *Madan et al., 2022*). To evaluate whether Fmi is also required to maintain a higher proliferative ratio in winners, we quantified mitotic cells in winner clones with and without Fmi. The wing disc displays distributed cell divisions throughout larval development, in contrast to eye discs, where passing of the morphogenetic furrow limits proliferation as cells start differentiating into compound eye cells (*Wolff and Ready, 1991*; *Tsachaki and Sprecher, 2012*). Therefore, for the quantification of proliferation, the wing is a better model than the eye disc. When wing disc >>Myc clones were depleted of Fmi, cell proliferation, as

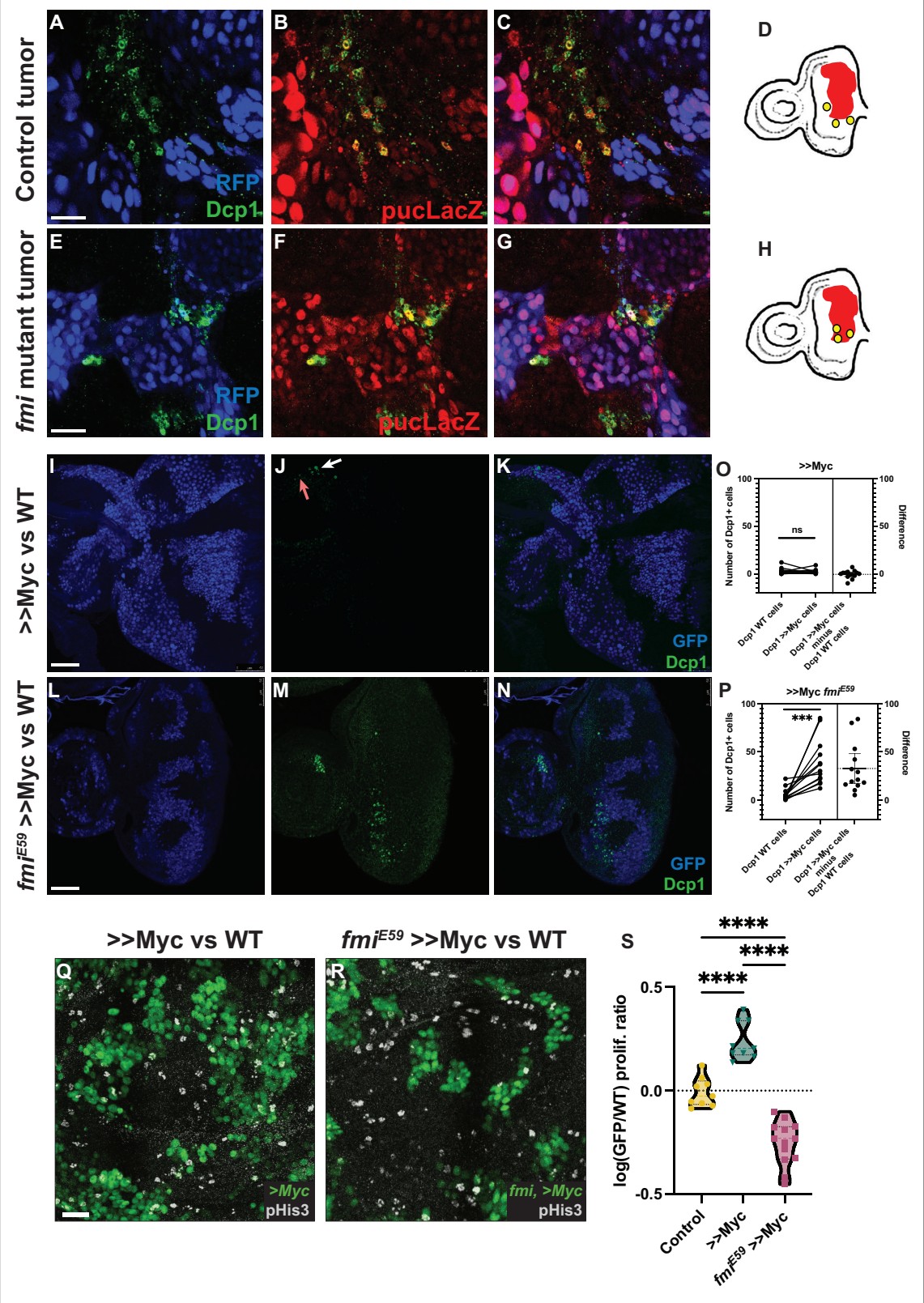

**Figure 4.** Lack of Fmi increases cell death and reduces proliferation in would-be winners. (**A–C**) Ras$^{V12}$, scrib RNAi tumors stained for DAPI, Dcp1+ (**A**), and puc-lacZ (**B**). (**C**) Merged channels. (**D**) Representation of how Dcp1+ staining localizes in the wildtype (WT) cells at the boundary with the tumor. (**E–G**) Ras$^{V12}$, scrib RNAi tumors mutant for Fmi, stained with DAPI, Dcp1+ (**E**), and puc-LacZ (**F**). (**G**) Merged channels. (**H**) Representation of how Dcp1+ staining localizes in the tumor cells in contact with the surrounding WT tissue. Scale bar for A–G: 25 µm. (**I–K**) Dcp1+ staining in >>Myc clones in eye

*Figure 4 continued*

discs. >>Myc clones are marked by GFP (**I**) and were stained against Dcp1+ (**J**). The arrows show apoptotic WT (red arrow) and >>Myc (white arrow) cells at the clone boundary. (**K**) Merged channels, showing apoptotic cells evenly distributed between WT and >>Myc cells. (**L–N**) Dcp1+ staining in >>Myc clones lacking Fmi in the eye disc. Eye disc >>Myc clones are marked by GFP (**L**) and were stained for Dcp1+ (**M, N**) Merged channels, showing apoptotic cells localized mainly in the >>Myc, *fmi$^{E59}$* clones. Scale bar: 50 μm. (**O**) Quantification of apoptotic cells in WT vs >>Myc clones in eye discs. Apoptosis occurs similarly in WT and >>Myc cells (two-tailed paired t-test; p-value = 0.6049). The left side of the graph shows the number of apoptotic WT and >>Myc cells. Each imaginal disc is displayed as a pair of dots, linked by a line, to easily visualize the Dcp1+ apoptotic cells in WT vs >Myc cells. Dots represent the number of apoptotic WT (left) or >>Myc (right) cells per disc. The right side of the graph displays the difference (>>Myc minus WT apoptotic cells). The dashed line indicates the mean difference between those values for all samples. N=14 discs. (**P**) Quantification of apoptotic cells in WT vs >>Myc, *fmi$^{E59}$* clones in eye discs. Apoptosis is found mainly in >>Myc, *fmi$^{E59}$* cells (two-tailed paired t-test; p-value = 0.0006). The left side of the graph shows the number of apoptotic WT and >>Myc, *fmi$^{E59}$* cells, side by side. Dots represent the number of apoptotic >>WT (left) or >>Myc, *fmi$^{E59}$* (right) cells per disc. The right side of the graph displays the difference (>>Myc, *fmi$^{E59}$* minus WT apoptotic cells). The dashed line indicates the mean difference between those values for all samples. N=14 discs. (**Q–R**) Proliferation analysis performed by pHis3 staining in wing discs with either >>Myc clones (**Q**) or >>Myc, *fmi$^{E59}$* clones (**R**). Scale bar: 20 μm. (**S**) Proliferative ratio of GFP cells in a non-competition Control (n=9 discs), >>Myc (n=9 discs), or >>Myc, *fmi$^{E59}$* (n=13 discs) clones. The proliferative ratio for each group was calculated as the ratio of pHis3 cells within the GFP+ clone vs the non-GFP WT tissue and the differences were analyzed as an ordinary ANOVA with a Tukey's test for multiple comparisons, with all p-values<0.0001 (****).

The online version of this article includes the following figure supplement(s) for figure 4:

**Figure supplement 1.** Fmi-/- tumor cell debris is found in vesicles inside wildtype cells.

**Figure supplement 2.** Lack of fmi does not affect the activation of JNK signaling.

measured by pHis3 staining, was significantly reduced (*Figure 4Q–S*). Taken together, these observations directly link the need for Fmi to proliferation and induced apoptosis, the two key events of cell competition, in several models of cell competition.

## Fmi-mediated cell-cell communication is not required for competition between winners and losers

Previously, some key players involved in cell competition, including Flower (*Rhiner et al., 2010*) and Xrp-1 (*Baillon et al., 2018*), were shown to be either up- or downregulated during competition. We evaluated whether Fmi protein levels might also be affected during competition. Fmi is ubiquitously expressed at low levels in *Drosophila* larva, pupa, and adult (*Brown et al., 2014*). We stained third instar larval wing discs with >>Myc clones to determine whether >>Myc supercompetition affects the levels of Fmi protein at the membrane, either inside the clone or near the boundary where cell competition occurs. We observed no change in Fmi protein levels (*Figure 5—figure supplement 1*), suggesting that Fmi is involved in competition through a mechanism that does not rely on altering protein levels.

Cell competition is thought to rely on intercellular communication to compare fitness and determine the outcome between prospective winner and loser cells (*Kon et al., 2017*; *Baker, 2020*; *Ogawa et al., 2021*). If Fmi is involved in these determinative intercellular communication events by signaling as a trans-homodimer, it should be required in both prospective winner and loser cells. Our previous eye and wing disc >>Myc supercompetition results already suggested this is not the case. While the removal of *fmi* caused winner clones to effectively become losers during >>Myc competition (*Figure 2*), removing *fmi* from the losers had no effect on the losers' outcome, either in eye discs (*Figure 2C and D*) or wing discs (*Figure 2G and H*). In both tissues, winner >>Myc clones showed no change in relative size to their twin spot counterparts, nor were loser clones eliminated more effectively (*Figure 2D and H*).

Our results so far suggested that the effect Fmi exerts on cell competition does not operate through bidirectional intercellular PCP communication. To further consolidate this hypothesis, we examined the effect of removing a dedicated core PCP protein other than Fmi from >>Myc supercompetitors. Generating *vang$^{A3}$* >>Myc clones in the wing disc, we observed that the >>Myc clones were unaffected and remained winners (*Figure 5*). *vang$^{A3}$*, >>Myc clones were on average 2.2±0.82 times larger than their hRFP twins, not significantly different from the 1.7±0.45-fold value for >>Myc clones (*Figure 2H*; *Figure 5C*).

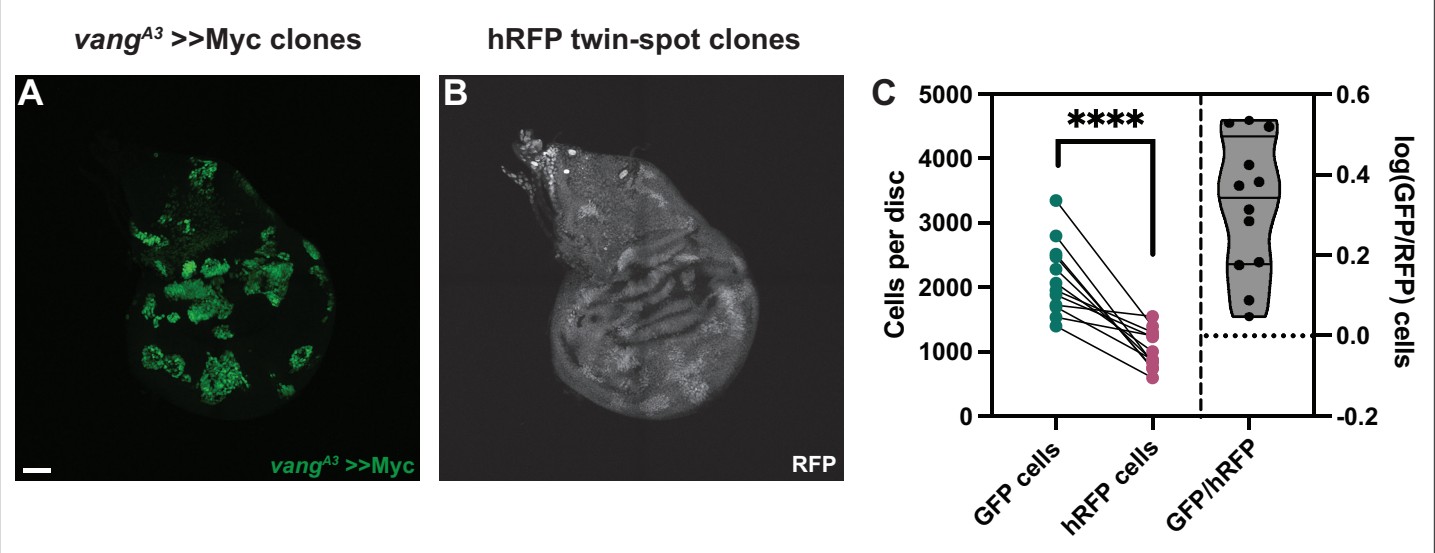

**Figure 5.** Vang is not required for >>Myc supercompetition. (**A**) Representative wing imaginal disc showing the GFP-tagged, *vang^A3* >>Myc clones. Clones were generated using hsp70-Flp. (**B**) RFP-labeled twin spots for the clones shown in A. Twin spot homozygous for RFP can be observed adjacent to the supercompetitor clones. (**C**) Graph showing the total GFP and homozygous RFP counts per disc, with each disc counts linked by a line and the log10 ratio of GFP/hRFP cells per disc (n=12 discs). Differences between GFP/hRFP clone size were analyzed using a two-tailed, ratio-paired t-test. p-Value<0.0001. When the ratio was compared with a two-tailed t-test to >>Myc, the difference was not significant, with a p-value of 0.078. Scale bar: 50 μm.

The online version of this article includes the following figure supplement(s) for figure 5:

**Figure supplement 1.** Fmi levels are not affected by cell competition.

## The activity of Fmi in cell competition does not require its cadherin domains

Our results have so far demonstrated that Fmi is required only in winner cells and that its function in cell competition is independent of PCP. Furthermore, our results rule out a role for Fmi homodimers in directly communicating fitness information between prospective winner and loser cells. However, the possibility remained that Fmi might function through trans-homodimerization between prospective winner cells to sustain winner status via a signal, via adhesion, or both. To ask if Fmi fulfills its role by mediating adhesion, we tested whether we could restore winner status to >>Myc clones that lack Fmi by providing a transgenic *fmi* that lacks the nine cadherin domains of Fmi (*arm-fmiΔCad*).

Co-expression of *arm-fmiΔCad* in *fmi^E59*, >>Myc clones re-established the ability of these clones to outcompete their neighbors, and they again became supercompetitors (*Figure 6*). When comparing the ratio of >>Myc clone cell number to twin spot cell number, the presence of *arm-fmiΔcad* restores competition to a level comparable to >>Myc supercompetitors (1.43±0.48 GFP/hRFP cells vs 1.71±0.45 GFP/hRFP cells respectively), far above the 0.49±0.29 value for *fmi^E59* >>Myc clones compared to twin spots.

Supercompetitor clones mutant for *fmi* consistently show clone fragmentation (*Figures 2F, 4R, and 6B*). Despite lacking the cadherin repeats, rescued clones not only restored their supercompetitor status, but fully recovered the ability of >>Myc clones to remain cohesive (*Figure 6C*). This suggests that the clone fragmentation we observed is unlikely due to the ability of Fmi to contribute to adhesion and more likely caused by a feature of the competition between wildtype cells and *fmi^-/-*>>Myc loser clones that causes their elimination.

Taken together, these results demonstrate that, while Fmi is essential in winner cells to eliminate less fit neighbors, this effect is independent of PCP or other homodimer-mediated signaling, and independent of Fmi-mediated cell adhesion, suggesting instead an as yet uncharacterized function for Fmi.

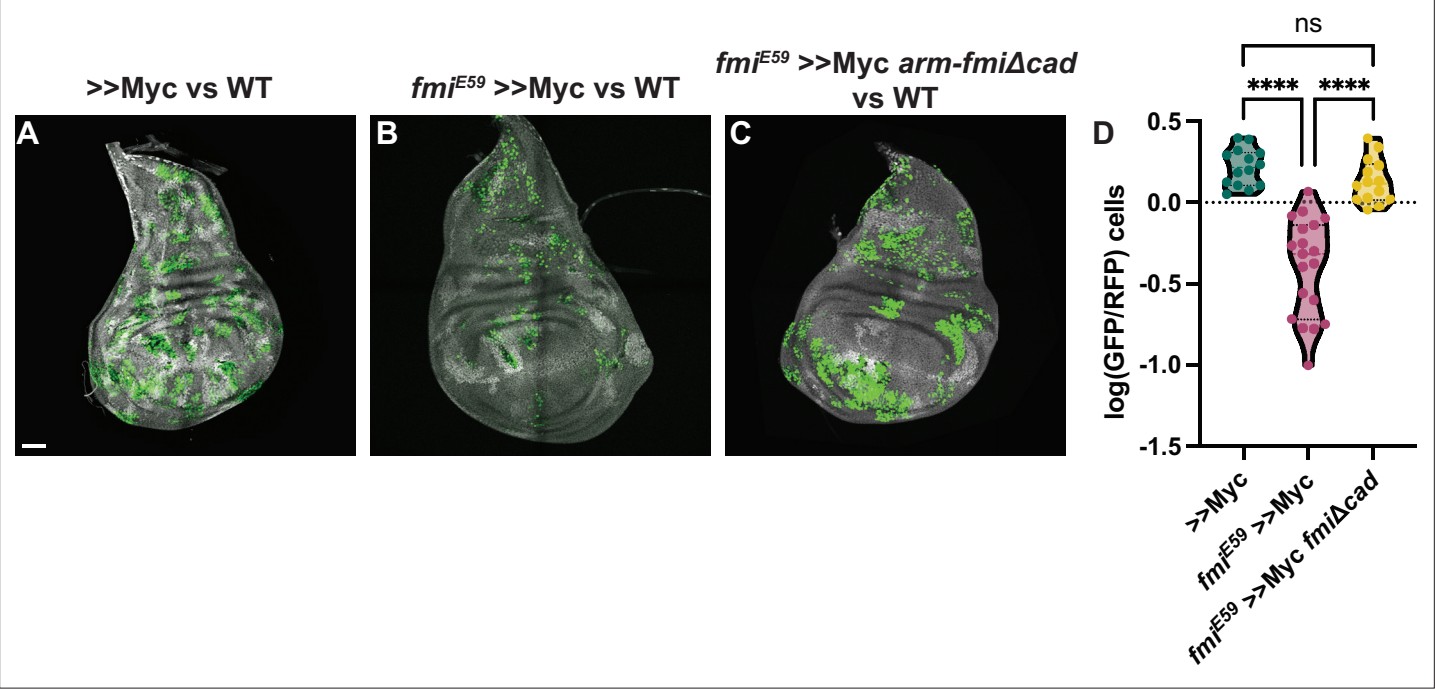

**Figure 6.** The Fmi cadherin repeats are not required for cell competition. (**A**) Representative disc with nGFP-labeled, >>Myc (**A**), >>Myc, *fmi^E59*
(**B**) or >>Myc, *fmi^E59*, *arm-fmiΔcad* (**C**) clones competing against wildtype twin spots in the wing disc. Twin spot clones are labeled with homozygous
nRFP and clones were generated with hsp70-Flp. (**D**) Ratio of GFP vs RFP cells in the three groups, represented as the log10(GFP/hRFP) cell ratios.
To evaluate the effect of the *arm-fmiΔcad* rescue (n=14 discs), the GFP/hRFP cell ratio was directly compared against the two other groups, already
quantified and shown in ***Figure 2F***. Differences between the groups were analyzed using an unpaired, ordinary one-way ANOVA, which found a p-
value<0.0001. Inter-group differences were analyzed with a Tukey's multiple comparisons test, which found no differences between >>Myc and >>Myc,
*fmi^E59*, *arm-fmiΔcad* clones, whereas both groups strongly differed from >>Myc, *fmi^E59* clones, both returning a p-value<0.0001 (****) when compared
directly against >>Myc, *fmi^E59*. Scale bar: 50 μm.

## Discussion

We have identified a requirement for Fmi in winner cells in both tumorigenic and developmental cell
competition models. Cells that would otherwise behave as winners instead behave as losers when
they lack Fmi. Fmi is notable in that it is required in cell competition in each of four distinct competi-
tion scenarios examined: Ras^V12 *scrib* RNAi tumors, Myc supercompetitor clones in eye and wing discs,
wildtype cells vs *scrib* RNAi loser clones in pupal eyes, and likely *scrib* RNAi clones in larval eye discs.
Just how universal this requirement is in other cell competition scenarios in *Drosophila* and perhaps in
competition in other organisms remains to be determined.

### Fmi acts in cell competition independently of PCP

Several arguments support the conclusion that the role for Fmi in cell competition is distinct from its
role in PCP signaling. First, Fmi is the only core PCP component among the six that were surveyed to
inhibit the ability of Ras^V12 *scrib* RNAi tumor clones to compete in the eye. If Fmi's role in cell compe-
tition were to signal winner fate to losers and vice versa by mirroring its role in PCP signaling, where
it signals the presence of proximal (Vang, Pk) components in one direction and the presence of distal
components (Fz, Dsh, Dgo) in the other direction between adjacent cells (*Lawrence et al., 2004*;
*Strutt and Strutt, 2007*; *Strutt and Strutt, 2008*; *Chen et al., 2008*; *Struhl et al., 2012*), then one
might expect either the proximal or distal components to also be required in winner clones. No such
requirement was observed. Second, in PCP signaling, Fmi functions as a trans-homodimer to transmit
those signals and requires the cadherin repeats and other extracellular domains, implying its func-
tion as a trans-homodimer (*Kimura et al., 2006*). In PCP signaling, removing Fmi from either of two
adjacent cells completely blocks PCP signaling. In contrast, in cell competition, while Fmi is required
in winners, removing Fmi from losers has no effect on competition. Thus, the model of bidirectional
signaling via Fmi is not supported by our results.

These observations, however, do not rule out the possibility that Fmi trans-homodimers contribute to intercellular signaling among winner cells. Nonetheless, as discussed above, adhesion seems not to be a meaningful part of the function for Fmi in winners, as an adhesion-deficient Fmi construct (*fmiΔCad*) fully rescues both competition and clone adhesion. The potential contributions of adhesion vs other possible signaling mechanisms are discussed at more length below.

Unlike the other core PCP genes, *fmi⁻/⁻* mutations are lethal due to requirements in the nervous system (***Usui et al., 1999***). Though not fully characterized, its roles in the nervous system appear to be distinct from PCP signaling. Fmi is required for outgrowth and guidance of the R8 axon of the eye to the M3 layer of the medulla via a mechanism that appears independent of other components of the core PCP signaling pathway (***Gao et al., 2000***; ***Lee et al., 2003***; ***Senti et al., 2003***), but does interact with Golden goal, a transmembrane phosphoprotein that is not associated with PCP signaling (***Takechi et al., 2021***). Growth of the dendrites of dorsal da neurons is also regulated by Fmi. During embryogenesis, da dendrites in fmi mutant embryos emerge precociously and overgrow as they approach the dorsal midline, and later, during larval growth, dendrites from opposite sides fail to avoid each other (tile) and instead overlap (***Gao et al., 2000***; ***Sweeney et al., 2002***; ***Kimura et al., 2006***).

Fmi is classified as an atypical cadherin and a Class-B adhesion G protein-coupled receptor (AGPCR), as it contains in its extracellular domain several conserved functional domains including cadherin repeats, epidermal growth factor-like repeats, laminin A G-type repeats, and a GPCR auto-proteolytic inducing (GAIN) domain that contains within it a GPCR proteolytic site (GPS) (***Rosa et al., 2021***; ***Einspahr and Tilley, 2022***; ***Sreepada et al., 2022***). These extracellular domains are followed by seven transmembrane domains and an intracellular C-terminal domain. Although much remains to be learned about this large subfamily of GPCRs, a general model has emerged in which activation by membrane bound or extracellular protein, peptide, proteoglycan or small molecule ligand, or mechanical force exposes a tethered ligand at the N-terminus of the C-terminal fragment in the GAIN domain that, upon exposure, interacts with the transmembrane portion to activate a G-protein signaling cascade. In many but not all AGPCRs, the tethered ligand is exposed by GAIN domain-mediated autoprotolysis of its GPS. Non-cleaved AGPCRs are hypothesized to expose the tethered ligand by an allosteric conformational change. Some de-orphanized AGPCRs interact with multiple ligands, and ligand binding can result in partial or full activation. In some cases, it appears that engineered truncation of portions of the extracellular domain can produce some level of ligand-independent activation (***Rosa et al., 2021***).

When expressed in da neurons, an Fmi construct lacking the cadherin, laminin G, and EGF-like repeats partially rescued the embryonic dendritic overgrowth phenotype (***Kimura et al., 2006***), suggesting a function independent of homodimerization. The G protein Gαq (Gq) has been proposed to function downstream of Fmi to mediate this repressive function (***Wang et al., 2016***). A recent preprint reports the resolved structure of the CELSR1 extracellular domain, showing that the protein has two distinct domains: an adhesion domain comprised of the first eight cadherin repeats, and a compact domain that extends from the ninth cadherin repeat to the transmembrane domains that is involved in GPCR signaling. Indeed, they demonstrated that a CELSR1 construct lacking only the cadherin repeats 1–8 retains the ability to activate Gαs, which has been predicted to interact with CELSR1 (***Bandekar et al., 2024***). Notably, our results showed that a similar Fmi construct lacking the cadherin domains substantially rescues the requirement for Fmi in winner cells during cell competition (***Figure 6***). These observations suggest that supplying adhesion is not the principal function of Fmi in these events, but are consistent with the possibility that homodimeric adhesion, or interaction with a different ligand, normally activates the receptor, and that the truncated FmiΔCad behaves as a constitutively activated receptor capable of binding to either Gα proteins. While no biochemical characterization of Fmi has been reported, the human orthologs CELSR1-3 have been studied in detail. CELSR2 is autoproteolytically cleaved while CELSRs 1 and 3 are not, yet all three couple to GαS (***Huong Bui et al., 2023***). Additional efforts will be required to determine the functional ligand(s) for Fmi and whether it signals similarly. Furthermore, AGPCRs participate in a wide variety of developmental and physiologic events through diverse effectors (***Einspahr and Tilley, 2022***; ***Sreepada et al., 2022***). The pathway by which Fmi participates in cell competition remains to be explored.

## Fmi, cell competition, and cancer

When the first examples of supercompetition were observed in Myc clones and the Hippo pathway (*Moreno and Basler, 2004*; *de la Cova et al., 2004*; *Ziosi et al., 2010*; *Neto-Silva et al., 2010*), they hinted at the possibility that tumors, which behave like supercompetitors, could use similar mechanisms to outcompete wildtype cells. Understanding tumor competition may open new avenues for early detection and therapy (*Baker and Li, 2008*; *Moreno, 2008*).

Research in *Drosophila* and mammals has shown that cell competition plays a dual role during tumorigenesis. Cells harboring mutations in proto-oncogenes or tumor-suppressor genes often behave as losers (*Maruyama and Fujita, 2017*; *Morata and Calleja, 2020*; *Kanda and Igaki, 2020*). Through the process of EDAC, epithelial tissues use cell competition to eliminate transformed pre-neoplastic cells by removing them from the tissue via directed cell death or extrusion (*Kon et al., 2017*; *Watanabe et al., 2018*). Pre-neoplastic cells that escape EDAC may accumulate additional mutations to become malignant tumors (*Watanabe et al., 2018*). Malignant tumors not only escape EDAC, but acquire properties that allow them to outcompete wildtype cells, facilitating invasion and metastasis (*Suijkerbuijk et al., 2016*; *Kohashi et al., 2021*). Furthermore, competition between clones within tumors further selects far more aggressive tumor behavior (*Parker et al., 2021*).

Another commonality between developmental and oncogenic cell competition is the involvement of the transmembrane protein Flower (Fwe). In both *Drosophila* and mammals, multiple isoforms of Fwe signal fitness; expression of Fwe$^{Lose}$ isoforms mark losers for elimination (*Rhiner et al., 2010*; *Merino et al., 2013*; *Levayer et al., 2015*; *Madan et al., 2019*). Forced expression of Fwe$^{Lose}$ induces cell competition and elimination of the loser, suggesting that Fwe comparison is involved in the sensing and/or initiation of differential fitness. This contrasts with Fmi, whose differential expression does not act as a trigger for competition, but which is needed in winners to allow them to win, suggesting that Fmi is involved after sensing in the execution of functions necessary to manifest winner behavior.

Evidence is accumulating that a human ortholog of Fmi, CELSR3, is expressed at high levels in a range of solid tumors, including lung, prostate, pancreatic, hepatic, ovarian, and colorectal cancers, and in some cases has been shown to be associated with poor prognosis (*Katoh and Katoh, 2007*; *Erkan et al., 2010*; *Asad et al., 2014*; *Goryca et al., 2018*; *Li et al., 2021*; *Chen et al., 2021c*). Recently, CELSR1 upregulation has also been linked to poor ovarian cancer prognosis, likely by promoting proliferation, migration, and invasion (*Zuo et al., 2023*). If CELSR1/3 are promoting winner cell behavior in these tumors, as might be predicted from its function in *Drosophila*, this could provide the rationale for future efforts to understand the mechanism by which Fmi/CELSR3 facilitates cell competition, with the goal of identifying an intervention that could blunt or perhaps even eliminate the aggressiveness of an array of highly morbid cancers.

# Materials and methods

### Key resources table

| Reagent type (species) or resource | Designation | Source or reference | Identifiers | Additional information |
|---|---|---|---|---|
| Gene (*Drosophila melanogaster*) | fmi | FlyBase | FBgn0024836 | Also known as stan |
| Gene (*D. melanogaster*) | fz | FlyBase | FBgn0001085 | |
| Gene (*D. melanogaster*) | dsh | FlyBase | FBgn0000499 | |
| Gene (*D. melanogaster*) | vang | FlyBase | FBgn0015838 | Also known as stbm |
| Gene (*D. melanogaster*) | pk | FlyBase | FBgn0003090 | |
| Gene (*D. melanogaster*) | dgo | FlyBase | FBgn0086898 | |
| Gene (*D. melanogaster*) | scrib | FlyBase | FBgn0263289 | |

*Continued on next page*

*Continued*

| Reagent type (species) or resource | Designation | Source or reference | Identifiers | Additional information |
|---|---|---|---|---|
| Gene (*D. melanogaster*) | myc | FlyBase | FBgn0262656 | |
| Gene (*D. melanogaster*) | Ras85D | FlyBase | FBgn0003205 | |
| Strain, strain background (*Escherichia coli*) | 5-alpha High Efficiency | NEB | C2987H | |
| Genetic reagent (*D. melanogaster*) | W RNAi | BDSC | 33623 | |
| Genetic reagent (*D. melanogaster*) | Fmi RNAi | BDSC | 26022 | |
| Genetic reagent (*D. melanogaster*) | Fz RNAi | BDSC | 31311 | |
| Genetic reagent (*D. melanogaster*) | Dsh RNAi | BDSC | 31306 | |
| Genetic reagent (*D. melanogaster*) | Vang RNAi | BDSC | 34354 | |
| Genetic reagent (*D. melanogaster*) | Pk RNAi | BDSC | 32413 | |
| Genetic reagent (*D. melanogaster*) | Scrib RNAi | BDSC | 39073 | |
| Transfected construct (*D. melanogaster*) | Arm-fmiΔCad | This paper | | Located in chromosome 2R |
| Antibody | Rabbit polyclonal α-pHis3 | Millipore | | 1:100 |
| Antibody | Rabbit monoclonal α-Dcp1 | Cell Signaling | RRID:AB_2721060 | 1:100 |
| Antibody | Mouse monoclonal α-LacZ | Promega | | 1:500 |
| Antibody | Rabbit polyclonal α-Cas3 | AbCam | | 1:200 |
| Antibody | 488-Goat polyclonal α-rabbit | Thermo Scientific | RRID:AB_3251385 | 1:500 |
| Antibody | 546-Goat polyclonal α-mouse | Thermo Scientific | RRID:AB_2535765 | 1:500 |
| Antibody | 546-Goat polyclonal α-rabbit | Thermo Scientific | RRID:AB_2534077 | 1:500 |
| Antibody | 647-Donkey polyclonal α-mouse | Thermo Scientific | RRID:AB_162542 | 1:500 |
| Commercial assay or kit | HiFi DNA assembly kit | New England Biolabs | | |
| Chemical compound, drug | Alexa 350 phalloidin | Thermo Scientific | | 1:500 |
| Chemical compound, drug | Alexa 635 phalloidin | Thermo Scientific | | 1:500 |
| Chemical compound, drug | DAPI | Invitrogen | | 1 µg/mL |
| Chemical compound, drug | Vectashield | Vector Labs | | |
| Software, algorithm | Fiji | https://fiji.sc | RRID:SCR_002285 | |
| Software, algorithm | Counting macros | https://github.com/iPabloSB/Nuclear-counts; **Shcherbina and Sanchez Bosch, 2023** | | |
| Software, algorithm | Prism 10 | GraphPad | RRID:SCR_002798 | |

## Resource availability

### Lead contact
Further information and requests for resources and reagents should be directed to and will be fulfilled by the lead contact (jaxelrod@stanford.edu).

### Materials availability
Plasmid and fly lines generated in this study are available from the lead contact upon request.

## Experimental model and study participant details

### Fly stocks and husbandry
Flies were maintained in standard fly food in a temperature-controlled incubator at either 25°C or 18°C. Egg collections were performed over a timespan of 24 or 48 hr. Vials with eggs were kept at 25°C until heat-shocked for clone generation or dissected.

Heat-shock was performed for 15 min on late first instar and early second instar larvae, 48 hr after egg laying (AEL). Vials were kept at 25°C after heat-shock until larvae were dissected.

The following fly lines were used for the experiments:

- y, w, hsp-Flp act5C-Gal4, UAS-nGFP to generate wing disc clones.
- ey-Flp; if /SM5^; TM2/^TM6b to generate eye disc clones and whole tumors.
- Bloomington TRiP UAS-RNAi lines RRID:BDSC_33623 (*w* control), RRID:BDSC_26022 (*fmi*), RRID:BDSC_31311 (*fz*), RRID:BDSC_31306 (*dsh*), RRID:BDSC_34354 (*vang*), RRID:BDSC_32413 (*pk*), RRID:BDSC_39073 (*scrib*).
- PCP mutant alleles FRT42D, *fmi$^{E45}$*/CyO, FRT42D, *fmi$^{E59}$*/CyO, FRT2A, *fz$^{R52}$*/TM6b, FRT42D, *vang$^{A3}$*, FRT42D *dgo$^{380}$*, FRT42D *pk$^{pk-sple13}$*.
- UAS-scrib RNAi; act5C>CD2>Gal4, UAS-RFP, UAS-Ras$^{V12}$/TM6b to make whole eye tumors.
- UAS-scrib RNAi FRT42D, tub-Gal$^{80}$; act5C>CD2>Gal4, UAS-RFP, UAS-Ras$^{V12}$/TM6b to generate tumor clones.
- UAS-scrib RNAi, FRT42D, tub-nRFP/FRT42D tub-Gal$^{80}$ to generate scrib RNAi eye disc clones.
- FRT42D *fmi$^{E59}$*/FRT42D, tub-nRFP, tub-Gal$^{80}$ to generate wing disc twin spots.
- UAS-dMyc/TM6b for eye and wing disc supercompetitor clones.
- FRT42D *fmi$^{E59}$*, arm-fmiΔCad to rescue wing disc clones.

## Genotypes of experimental models

Figure 1:

(B) ey-Flp; act5C>CD2>Gal4, UAS-nRFP / TM6b.

(C–H) ey-Flp; UAS-scrib RNAi / UAS-DCR2; act5C>CD2>Gal4, UAS-nRFP, UAS-Ras$^{V12}$/TM6b crossed to the homozygous UAS-RNAi line noted above.

(J) ey-Flp; FRT42D tub-Gal$^{80}$/FRT42D; act5C>CD2>Gal4, UAS-nRFP/TM6b.

(K) ey-Flp; FRT42D tub-Gal$^{80}$/FRT42D; act5C>CD2>Gal4, UAS-nRFP UAS-Ras$^{V12}$/UAS-*scrib* RNAi.

(L) ey-Flp; FRT42D tub-Gal$^{80}$/FRT42D *fmi$^{E45}$*; act5C>CD2>Gal4, UAS-nRFP UAS-Ras$^{V12}$/UAS-*scrib* RNAi.

(M) ey-Flp; act5C>CD2>Gal4, UAS-nRFP UAS-Ras$^{V12}$/UAS-*scrib* RNAi; *fz$^{R52}$* FRT2A/tub-Gal$^{80}$ FRT2A.

(N) ey-Flp; FRT42D tub-Gal$^{80}$/FRT42D *dgo$^{380}$*; act5C>CD2>Gal4, UAS-nRFP UAS-Ras$^{V12}$/UAS-*scrib* RNAi.

(O) ey-Flp; FRT42D tub-Gal$^{80}$/FRT42D *pk$^{pk-sple13}$*; act5C>CD2>Gal4, UAS-nRFP UAS-Ras$^{V12}$/UAS-*scrib* RNAi.

(P) ey-Flp; FRT42D tub-Gal$^{80}$/FRT42D *vang$^{A3}$*; act5C>CD2>Gal4, UAS-nRFP UAS-Ras$^{V12}$/UAS-*scrib* RNAi.

Figure 2:

(A) ey-Flp; FRT42D tub-Gal$^{80}$/FRT42D; act5C>CD2>Gal4 UAS-nRFP/UAS-dMyc.

(B) ey-Flp; FRT42D tub-Gal$^{80}$/FRT42D *fmi$^{E59}$*; act5C>CD2>Gal4 UAS-nRFP/UAS-dMyc.

(C) ey-Flp; FRT42D *fmi$^{E59}$* tub-Gal$^{80}$/FRT42D tub-nRFP; act5C>CD2>Gal4 UAS-nGFP/UAS-dMyc.

(E) hsp-Flp tub-Gal4, UAS-nGFP; FRT42D tub-nRFP tub-Gal$^{80}$/FRT42D; UAS-dMyc / +.
(F) hsp-Flp tub-Gal4, UAS-nGFP; FRT42D tub-nRFP tub-Gal$^{80}$/FRT42D *fmi$^{E59}$*; UAS-dMyc / +.
(G) hsp-Flp tub-Gal4, UAS-nGFP; FRT42D *fmi$^{E59}$* tub-nRFP tub-Gal$^{80}$/FRT42D; UAS-dMyc / +.

Figure 3:

(A–C) ey-Flp; FRT42D tub-Gal80/FRT42D; act5C>CD2>Gal4 UAS-nRFP / +.
(D–F) ey-Flp; FRT42D tub-Gal80/FRT42D; act5C>CD2>Gal4 UAS-nRFP/UAS-*scrib* RNAi *puc$^{E69}$*.
(G–I) ey-Flp; FRT42D tub-Gal80/FRT42D *fmi$^{E59}$*; act5C>CD2>Gal4 UAS-nRFP/UAS-*scrib* RNAi *puc$^{E69}$*.
(J–K) ey-Flp; FRT42D *fmi$^{E59}$* tub-Gal80/FRT42D; act5C>CD2>Gal4 UAS-nRFP/UAS-*scrib* RNAi *puc$^{E69}$*.

Figure 4:

(A–C) ey-Flp; UAS-*scrib* RNAi FRT42D tub-Gal$^{80}$/FRT42D; act5C>CD2>Gal4, UAS-nRFP UAS-Ras$^{V12}$/*pucE$^{E69}$*.
(E–G) ey-Flp; UAS-*scrib* RNAi FRT42D tub-Gal$^{80}$/FRT42D *fmi$^{E59}$*; act5C>CD2>Gal4, UAS-nRFP UAS-Ras$^{V12}$/*pucE$^{E69}$*.
(I–K) ey-Flp; FRT42D tub-Gal$^{80}$/FRT42D *fmi$^{E59}$*; act5C>CD2>Gal4 UAS-nRFP/UAS-dMyc.
(L) hsp-Flp tub-Gal4, UAS-nGFP; FRT42D tub-nRFP tub-Gal$^{80}$/FRT42D; UAS-dMyc / +.
(M) hsp-Flp tub-Gal4, UAS-nGFP; FRT42D tub-nRFP tub-Gal$^{80}$/FRT42D *fmi$^{E59}$*; UAS-dMyc / +.

Figure 5:

(A, B) hsp-Flp tub-Gal4, UAS-nGFP; FRT42D tub-nRFP tub-Gal$^{80}$/FRT42D *vang$^{A3}$*; UAS-dMyc / +.

Figure 6:

(A) hsp-Flp tub-Gal4, UAS-nGFP; FRT42D tub-nRFP tub-Gal$^{80}$/FRT42D; UAS-dMyc / +.
(B) hsp-Flp tub-Gal4, UAS-nGFP; FRT42D tub-nRFP tub-Gal$^{80}$/FRT42D *fmi$^{E59}$*; UAS-dMyc / +.
(C) hsp-Flp tub-Gal4, UAS-nGFP; FRT42D tub-nRFP tub-Gal$^{80}$/FRT42D *fmi$^{E59}$* arm-*fmiΔCad*; UAS-dMyc / +.

Figure 1—figure supplement 1:

(A, B) ey-Flp; FRT42D tub-Gal$^{80}$/FRT42D tub-nRFP; act5C>CD2>Gal4/UAS-nGFP.

Figure 1—figure supplement 2:

(A) ey-Flp; *scrib* RNAi FRT42D tub-Gal$^{80}$/FRT42D; act5C>CD2>Gal4, UAS-nRFP, UAS-Ras$^{V12}$/*w* RNAi.
(B) ey-Flp; *scrib* RNAi FRT42D tub-Gal$^{80}$/FRT42D; act5C>CD2>Gal4, UAS-nRFP, UAS-Ras$^{V12}$/*fmi* RNAi.
(C) ey-Flp; FRT42D / FRT42D GMR-Hid, l(2)CL-R; act5C>CD2>Gal4, UAS-nRFP, UAS-Ras$^{V12}$/*scrib* RNAi.
(D) ey-Flp; FRT42D *fmi$^{E59}$*/FRT42D GMR-Hid, l(2)CL-R; act5C>CD2>Gal4, UAS-nRFP, UAS-Ras$^{V12}$/*scrib* RNAi.

Figure 2—figure supplement 1:

(A) ey-Flp; FRT42D tub-Gal$^{80}$/FRT42D; act5C>CD2>Gal4/UAS-nRFP.
(B) ey-Flp; FRT42D tub-Gal$^{80}$/FRT42D *fmi$^{E59}$*; act5C>CD2>Gal4/UAS-nRFP.
(D, E) hsp-Flp tub-Gal4, UAS-nGFP; FRT42D tub-nRFP tub-Gal$^{80}$/FRT42D *fmi$^{E59}$*; MKRS / TM6b.

Figure 4—figure supplement 1:

(A–C) ey-Flp; FRT42D tub-Gal$^{80}$/FRT42D *fmi$^{E45}$*; act5C>CD2>Gal4, UAS-nRFP UAS-Ras$^{V12}$/UAS-*scrib* RNAi.

Figure 4—figure supplement 2:

(A–C) ey-Flp; UAS-*scrib* RNAi FRT42D tub-Gal$^{80}$/FRT42D; act5C>CD2>Gal4, UAS-nRFP UAS-Ras$^{V12}$/*pucE$^{E69}$*.

(D–F) ey-Flp; UAS-*scrib* RNAi FRT42D tub-Gal[80]/FRT42D *fmi*[E59]; act5C>CD2>Gal4, UAS-nRFP UAS-Ras[V12]/*pucE*[E69].

Figure 5—figure supplement 1:

(A–F) hsp-Flp, tub-Gal4, UAS-nGFP; FRT42D tub-Gal[80]/FRT42D; UAS-dMyc / +.

## Method details

### Generation of arm-fmiΔCad

To make the *arm-fmiΔCad* construct, we used cDNA from the *fmi isoform A (stan-RA)* terminally tagged with an HA tag, fused using an SGGGGS linker (*fmi::HA*). We subcloned by Gibson assembly a PCR fragment containing the coding sequence for *fmi::HA* lacking the first 1328 aa, which contain the 9 cadherin domains (fmi$^{\Delta 1\text{-}1328}$) into a pCaSpeR4 vector backbone with the *armadillo* promoter and an Fz 5'UTR. The pCaSpeR4-armP-*fmiΔCad* construct was introduced into flies by P-element integration, and we used a fly line carrying the construct on chromosome arm 2R.

### Wing imaginal disc dissection and immunohistochemistry

Third instar wandering larvae (120 hr AEL) were dissected by transversally cutting the larva in two halves. The posterior half was discarded, and the anterior part was inverted, after which the fat body and digestive tissue (mouth hooks, salivary glands, and gut) were removed. Inverted larvae were fixed in 4% paraformaldehyde (PFA) in phosphate-buffered saline (PBS)+0.02% Triton X-100 (PBS-T) for 30 min at room temperature (RT). Fixed larvae were then washed three times with PBS-T and then stained.

For antibody staining, larvae were stained with primary antibodies diluted in PBS-T+3% normal donkey serum (NDS) overnight at 4°C, and then stained with secondary antibodies and DAPI for 1 hr at room temperature. Stained larvae were then washed three times with PBS. Wing discs were carefully removed from the inverted larva and mounted in Vectashield mounting medium (Vector Labs).

We used the following primary antibodies: rabbit α-pHis3 (Millipore), 1:100; rabbit α-Dcp1 (Cell Signaling), 1:100. Secondary staining was performed with Thermo Scientific 546-goat α-rabbit, 1:500.

### Immunohistochemistry of third instar larval eye discs

Discs dissected from late third instar larvae were fixed for 5–15 min in 4% PFA in PBS at 4°C. Fixed eye discs were washed two times in PBS-T. After blocking for 1 hr in 5% bovine serum albumin in PBS-T at 4°C, discs were incubated with primary antibodies in the blocking solution overnight at 4°C. Incubations with secondary antibodies were done for 90 min at room temperature in PBS-T. Secondary antibody was washed three times with PBS-T. Incubations in phalloidin (1:200) and DAPI (1 µg/mL), if required, were done in PBS-T for 15 min followed by washing at room temperature before mounting. Stained samples were mounted in 15 µL Vectashield mounting medium (Vector Labs). We used the following primary antibodies: mouse anti-LacZ (1:500 dilution, Promega) rabbit α-Dcp1 (1:500 dilution, Cell Signaling), mouse α-Fmi (1:200 dilution, DSHB). We used the following secondary antibodies from Thermo Scientific: 488-goat α-rabbit, 546-goat α-mouse, 594-donkey α-mouse, 647-donkey α-mouse, Alexa 635- and Alexa 350-conjugated phalloidin.

### Image acquisition

Images of whole discs were taken with a Leica SP8 system equipped with a White Light Laser and HyD detectors. A ×40, NA 1.5 Leica objective and 1.51 refractive index immersion oil were used. Image stacks were taken in 8-bit at 1024×1024 px resolution, and a pinhole of 1 airy unit (AU), using a z-step of 0.3 µM. For automated image quantification pipelines, residual tissue from the leg or haltere discs was removed in Fiji (fiji.sc, *Schindelin et al., 2012*), as well as the peripodial cells at the apical section of the Z-stack.

Adult eyes and RFP/GFP signal from pupae and eye discs isolated from late third instar larvae were imaged on a Leica MZ16F Stereomicroscope.

## Quantification and statistical analysis

### Eye disc clone analysis

Eye imaginal discs representative slices were selected to measure the size of RFP+ clones. To do so, the GFP+ clone area was divided by the non-fluorescent eye disc area to obtain the ratio of GFP+ clone vs non-GFP twin and plotted as the log10(ratio) on violin plots including all data points, median, and quartiles. Statistical analysis was performed in GraphPad Prism 10. Data was analyzed using unpaired, two-tailed t-tests.

### Wing disc clone analysis

Wing imaginal disc GFP clone and RFP twin spot cell counts were obtained using a set of automated macros written for ImageJ (*Sanchez Bosch and Axelrod, 2024*). Cell ratios were obtained as the fraction of GFP+ cells vs RFP+ twin spot cells. Cell ratios were transformed as the log10(ratio) and graphed as violin plots including all data points, median, and quartiles. Statistical analysis was performed in GraphPad Prism 10. Data was analyzed using either unpaired, two-tailed t-test (*Figure 5*), or an ordinary one-way ANOVA with a Tukey's multiple comparisons test (*Figures 2–4 and 6*).

### Apoptosis quantification

Apoptotic cells marked with positive Dcp1 staining were scored when located one to three cells away from a clone boundary, as those were arbitrarily deemed as caused by cell competition. Cell counts were plotted individually, with each disc's WT and GFP cells plotted side-by-side, linked by a straight line. The difference in apoptosis between WT and >>Myc (or >>Myc, $fmi^{E59}$), GFP+ cells in each disc was then calculated (GFP minus WT cells) and plotted on the right. The mean difference of the analysis was also plotted in the same graph as a dashed line. The statistical differences were obtained using a paired, two-tailed t-test.

### Cell proliferation quantification

Clone proliferation ratios were measured in ImageJ. First, we obtained the number of GFP+ cells vs the cells outside the GFP+ clones by using the same macros used to quantify GFP clones and then counting the total cells in the disc by counting the DAPI nuclei (*Sanchez Bosch and Axelrod, 2024*), and then subtracting the GFP+ cells from the total DAPI cell counts. Then, pHis3-positive cells were located by using the 3D Find maxima function from the ImageJ 3D Suite (https://mcib3d.frama.io/3d-suite-imagej/). To do so, the pHis3 channel was processed as follows: (1) specks were removed by using the Remove Outliers (radius of 5, threshold of 50), then the background was removed with a 2$Px$ 3D Gaussian Blur and the Subtract background function (rolling ball radius = 10 px, with sliding parabolic and disabled smoothing). Last, the pHis3+ peaks were found using the 3D Maxima finder (minimum threshold of 5 px, with a XY and Z radius of 3 px and discarding all peaks below the noise level of 20).

GFP+, pHis3 cells were obtained by first creating a binary mask of the GFP channel by smoothing the image with a 5 px 3D Gaussian blur and then using a Li threshold to create the mask. To ensure proper quantification, correct thresholding of the clones was visually assessed and adjusted when needed. Then, all pHis3 peaks that were inside the GFP+ clone were counted as proliferative GFP+ cells. Last the proliferative ratio of GFP+ cells was obtained by dividing the fraction of pHis3 cells inside GFP clones by the fraction of pHis3 cells outside of GFP+ clones. The log10 of the proliferative ratio was plotted as a violin plot representing each disc as a data point and indicating the median and quartiles. Statistical analysis was performed in GraphPad Prism 10. Data was analyzed using an ordinary one-way ANOVA and the groups were compared to each other using a Tukey's multiple comparisons test.

## Acknowledgements

We would like to thank members of the Axelrod lab for fruitful discussions. Stocks obtained from the Bloomington *Drosophila* Stock Center (NIH P40OD018537) were used in this study. This work was funded by NIH R35GM131914 (JDA) and the Swiss National Science Foundation P400PB_199258 (PSB). The funders had no role in study design, data collection and analysis, decision to publish, or preparation of the manuscript.

## Additional information

### Funding

| Funder | Grant reference number | Author |
|---|---|---|
| National Institutes of Health | R35GM131914 | Jeffrey D Axelrod |
| Swiss National Science Foundation | P400PB_199258 | Pablo Sanchez Bosch |

The funders had no role in study design, data collection and interpretation, or the decision to submit the work for publication.

### Author contributions

Pablo Sanchez Bosch, Conceptualization, Software, Formal analysis, Funding acquisition, Methodology, Writing – original draft, Writing – review and editing; Bomsoo Cho, Conceptualization, Formal analysis, Investigation; Jeffrey D Axelrod, Conceptualization, Supervision, Funding acquisition, Writing – original draft, Project administration, Writing – review and editing

### Author ORCIDs

Pablo Sanchez Bosch ⑩ https://orcid.org/0000-0002-0574-4530
Bomsoo Cho ⑩ https://orcid.org/0000-0001-8970-9160
Jeffrey D Axelrod ⑩ https://orcid.org/0000-0001-6094-7392

Reviewer #1 (Public review): https://doi.org/10.7554/eLife.98535.4.sa1
Reviewer #2 (Public review): https://doi.org/10.7554/eLife.98535.4.sa2
Reviewer #3 (Public review): https://doi.org/10.7554/eLife.98535.4.sa3
Author response https://doi.org/10.7554/eLife.98535.4.sa4

## Additional files

### Supplementary files

MDAR checklist

Supplementary file 1. Source data for figure and figure supplements.

### Data availability

All images acquired for the study are available in the BioStudies database under accession number S-BIAD1513. The code for the ImageJ/Fiji macros used in this study can be accessed and downloaded from GitHub (copy archived at *Shcherbina and Sanchez Bosch, 2023*). Numerical data for all graphs in this study is provided in *Supplementary file 1*. Any additional information required to reanalyze the data reported in this paper is available from the lead contact upon request.

The following dataset was generated:

| Author(s) | Year | Dataset title | Dataset URL | Database and Identifier |
|---|---|---|---|---|
| Bosch PS, Cho B, Axelrod JD | 2024 | Flamingo participates in multiple models of cell competition | https://www.ebi.ac.uk/biostudies/bioimages/studies/S-BIAD1513 | ArrayExpress, S-BIAD1513 |

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
