## [Editor Report · eLife Assessment]

This study investigates the role of the Cadherin Flamingo (Fmi) in cell competition in developing tissues in *Drosophila melanogaster*. The findings are **valuable** in that they show that Fmi is required in winning cells in several competitive contexts. The evidence supporting the conclusions is **solid**, as the authors identify Fmi as a potential new regulator of cell competition, however, they don't delve into a mechanistic understanding of how this occurs.

---

## [Referee Report · Reviewer #1 (Public review)]

Summary:

This paper is focused on the role of Cadherin Flamingo (Fmi) in cell competition in developing *Drosophila* tissues. A primary genetic tool is monitoring tissue overgrowths caused by making clones in the eye disc that expression activated Ras (RasV12) and that are depleted for the polarity gene scribble (scrib). The main system that they use is ey-flp, which make continuous clones in the developing eye-antennal disc beginning at the earliest stages of disc development. It should be noted that RasV12, scrib-i (or lgl-i) clones only lead to tumors/overgrowths when generated by continuous clones, which presumably creates a privileged environment that insulates them from competition. Discrete (hs-flp) RasV12, lgl-i clones are in fact out-competed (PMID: 20679206), which is something to bear in mind. They assess the role of fmi in several kinds of winners, and their data support the conclusion that fmi is required for winner status. However, they make the claim that loss of fmi from Myc winners converts them to losers, and the data supporting this conclusion is not compelling.

Strengths:

Fmi has been studied for its role in planar cell polarity, and its potential role in competition is interesting.

---

## [Referee Report · Reviewer #2 (Public review)]

Summary:

In this manuscript, Bosch et al. reveal Flamingo (Fmi), a planar cell polarity (PCP) protein, is essential for maintaining 'winner' cells in cell competition, using *Drosophila* imaginal epithelia as a model. They argue that tumor growth induced by scrib-RNAi and RasV12 competition is slowed by Fmi depletion. This effect is unique to Fmi, not seen with other PCP proteins. Additional cell competition models are applied to further confirm Fmi's role in 'winner' cells. The authors also show that Fmi's role in cell competition is separate from its function in PCP formation.

Strengths:

(1) The identification of Fmi as a potential regulator of cell competition under various conditions is interesting.

(2) The authors demonstrate that the involvement of Fmi in cell competition is distinct from its role in planar cell polarity (PCP) development.

---

## [Referee Report · Reviewer #3 (Public review)]

Summary:

In this manuscript, Bosch and colleagues describe an unexpected function of Flamingo, a core component of the planar cell polarity pathway, in cell competition in *Drosophila* wing and eye disc. While Flamingo depletion has no impact on tumour growth (upon induction of Ras and depletion of Scribble throughout the eye disc), and no impact when depleted in WT cells, it specifically tunes down winner clone expansion in various genetic contexts, including the overexpression of Myc, the combination of Scribble depletion with activation of Ras in clones or the early clonal depletion of Scribble in eye disc. Flamingo depletion reduces proliferation rate and increases the rate of apoptosis in the winner clones, hence reducing their competitiveness up to forcing their full elimination (hence becoming now "loser"). This function of Flamingo in cell competition is specific of Flamingo as it cannot be recapitulated with other components of the PCP pathway, does not rely on interaction of Flamingo in trans, nor on the presence of its cadherin domain. Thus, this function is likely to rely on a non-canonical function of Flamingo which may rely on downstream GPCR signaling.

This unexpected function of Flamingo is by itself very interesting. In the framework of cell competition, these results are also important as they describe, to my knowledge, one of the only genetic conditions that specifically affect the winner cells without any impact when depleted in the loser cells. Moreover, Flamingo do not just suppress the competitive advantage of winner clones, but even turn them in putative losers. This specificity, while not clearly understood at this stage, opens a lot of exciting mechanistic questions, but also a very interesting long term avenue for therapeutic purpose as targeting Flamingo should then affect very specifically the putative winner/oncogenic clones without any impact in WT cells.

The data and the demonstration are very clean and compelling, with all the appropriate controls, proper quantifications and backed-up by observations in various tissues and genetic backgrounds. I don't see any weakness in the demonstration and all the points raised and claimed by the authors are all very well substantiated by the data. As such, I don't have any suggestions to reinforce the demonstration.

While not necessary for the demonstration, documenting the subcellular localisation and levels of Flamingo in these different competition scenarios may have been relevant and provide some hints on a putative mechanism (specifically by comparing its localisation in winner and loser cells).

Also, on a more interpretative note, the absence of impact of Flamingo depletion on JNK activation does not exclude some interesting genetic interactions. JNK output can be very contextual (for instance depending on Hippo pathway status), and it would be interesting in the future to check if Flamingo depletion could somehow alter the effect of JNK in the winner cells and promote downstream activation of apoptosis (which might normally be suppressed). It would be interesting to check if Flamingo depletion could have an impact in other contexts involving JNK activation or upon mild activation of JNK in clones.

Strengths:

- A clean and compelling demonstration of the function of Flamingo in winner cells during cell competition

- One of the rare genetic conditions that affects very specifically winner cells without any impact in losers, and then can completely switch the outcome of competition (which opens an interesting therapeutic perspective on the long term)

---

## [Author Response]

The following is the authors’ response to the previous reviews.

(1) We agreed that there was insufficient evidence for the authors' conclusion that Myc-overexpressing clones lacking Fmi become losers. We request that the authors change the text to discuss that suppression of Myc clone growth through Fmi depletion is reminiscent of a cell acquiring loser status, although at this point in the manuscript there is no clear demonstration whether this is mostly driven by growth suppression and/or an increase in apoptosis.

We agree that at the point in the manuscript where we have only described the clone sizes, one cannot make firm conclusions about competition, so we have changed the language to reflect this. We argue that after showing our apoptosis data, those conclusions become firm. Please see the more lengthy responses to reviewers below.

(2) We agreed that the apoptosis assay, data and interpretation need to be improved. The graphs in Fig. 4O and P should be better discussed in the text and in the legend. Additionally, the graphs are lacking the red lines that are written in the text.

We regret that we did not adequately explain the data displayed in these two graphs. Supercompetition tends to cause apoptosis in both winners and losers, with the ratio between WT and super-competitor cells being critical in deciding the outcome of competition. We wanted to represent this visually but failed to properly explain our analysis. We have rewritten the figure legend and our discussion in the main text, hopefully making it clearer.

**Public Reviews:**

**Reviewer #1 (Public review):**
Summary:This paper is focused on the role of Cadherin Flamingo (Fmi) in cell competition in developing *Drosophila* tissues. A primary genetic tool is monitoring tissue overgrowths caused by making clones in the eye disc that expression activated Ras (RasV12) and that are depleted for the polarity gene scribble (scrib). The main system that they use is ey-flp, which make continuous clones in the developing eye-antennal disc beginning at the earliest stages of disc development. It should be noted that RasV12, scrib-i (or lgl-i) clones only lead to tumors/overgrowths when generated by continuous clones, which presumably creates a privileged environment that insulates them from competition. Discrete (hs-flp) RasV12, lgl-i clones are in fact out-competed (PMID: 20679206), which is something to bear in mind. They assess the role of fmi in several kinds of winners, and their data support the conclusion that fmi is required for winner status. However, they make the claim that loss of fmi from Myc winners converts them to losers, and the data supporting this conclusion is not compelling.Strengths:Fmi has been studied for its role in planar cell polarity, and its potential role in competition is interesting.Weaknesses:I have read the revised manuscript and have found issues that need to be resolved. The biggest concern is the overstatement of the results that loss of fmi from Myc-overexpressing clones turns them into losers. This is not shown in a compelling manner in the revised manuscript and the authors need to tone down their language or perform more experiments to support their claims. Additionally, the data about apoptosis is not sufficiently explained.

We take issue with this reviewer’s framing of their criticism. First, the reviewer is selectively reporting the results published in PMID: 20679206. They correctly state that those authors show that small discreet clones of RasV12 lgl are eliminated (Fig. 3B), but they omit the fact that the authors also show that larger RasV12 lgl clones induce apoptosis in the surrounding wild type cells, and therefore behave as winners (Fig. 3C). Hence, the size of the clone appears to determine its winner/loser status. Of course, lgl is not scrib, and it is not a certainty that they would behave similarly, but they also show that large RasV12 scrib clones induce considerable apoptosis of the neighboring wild type cells.

The reviewer then discusses “continuous” clones induced by ey-flp, as we use in our manuscript. Here, the term “continuous” is probably misleading; because ey is expressed ubiquitously in the disc from early in development, it is most likely the case that the majority of cells have flipped relatively early, resulting in ~half the cells becoming clone and the other ~half twin spot. The clone cells then likely fuse to make larger clones. We show that ey-flp induced RasV12 scrib clones also behave as winners. It is logical to conclude that this is because they are large. The reviewer talks about “a privileged environment that insulates them from competition,” but if they were insulated from competition, how could they become winners? Because they occupy more territory than the wild type cells, and because they induce apoptosis in the wild type neighbors, they are winners.

Having shown that ey-flp induced RasV12 scrib clones behave as winners, we then remove Fmi from these clones, and show that they behave as losers by the same criteria: they occupy less area than the wild type cells (our Fig. 1 and Fig. 1 Supp 2), and they induce apoptosis in the wild type cells (our Fig 4A-H).

With respect to the comment about additional experiments are needed to support the claim that loss of Fmi from Myc winners converts them to losers, we’re not sure what additional data the reviewer would want. As for the tumor clones, we show that >>Myc clones get bigger than the twin control clones (Fig. 2), and we measure similar low levels of apoptosis in each (Fig. 4I-K, O). In contrast >>Myc fmi clones are out-grown by wild type clones, and apoptosis is higher in the >>Myc fmi clones than in the wild type clones (Fig. 4L-N, P-S). We therefore believe it is correct to say that >>Myc clones become losers when Fmi is removed.

In additional comments, the reviewer takes issue with using winner and loser language at the point in the manuscript where we have only shown the clone sizes but not yet the apoptosis data, and about this we agree. We have changed the language accordingly.

Re explanation of the apoptosis data, see the response to reviewer #3.

**Reviewer #2 (Public review):**
Summary:In this manuscript, Bosch et al. reveal Flamingo (Fmi), a planar cell polarity (PCP) protein, is essential for maintaining 'winner' cells in cell competition, using *Drosophila* imaginal epithelia as a model. They argue that tumor growth induced by scrib-RNAi and RasV12 competition is slowed by Fmi depletion. This effect is unique to Fmi, not seen with other PCP proteins. Additional cell competition models are applied to further confirm Fmi's role in 'winner' cells. The authors also show that Fmi's role in cell competition is separate from its function in PCP formation.Strengths:(1) The identification of Fmi as a potential regulator of cell competition under various conditions is interesting.(2) The authors demonstrate that the involvement of Fmi in cell competition is distinct from its role in planar cell polarity (PCP) development.Weaknesses:(1) The authors provide a superficial description of the related phenotypes, lacking a mechanistic understanding of how Fmi regulates cell competition. While induction of apoptosis and JNK activation are commonly observed outcomes in various cell competition conditions, it is crucial to determine the specific mechanisms through which they are induced in fmi-depleted clones. Furthermore, it is recommended that the authors utilize the power of fly genetics to conduct a series of genetic epistasis analyses.

We agree that it is desirable to have a mechanistic understanding of Fmi’s role in competition, but that is beyond the scope of this manuscript. Here, our goal is to report the phenomenon. We understand and share with the reviewer the interest in better understanding the relationship between Fmi and JNK signaling in competition. The role of JNK in competition, tumorigenesis and cell death is infamously complex. In some preliminary experiments, we explored some epistasis experiments, but these were inconclusive so we elected to not report them here. In the future, we will continue with additional analyses to gain a better understanding of the mechanism by which Fmi affects competition.

**Reviewer #3 (Public review):**
Summary:In this manuscript, Bosch and colleagues describe an unexpected function of Flamingo, a core component of the planar cell polarity pathway, in cell competition in *Drosophila* wing and eye disc. While Flamingo depletion has no impact on tumour growth (upon induction of Ras and depletion of Scribble throughout the eye disc), and no impact when depleted in WT cells, it specifically tunes down winner clone expansion in various genetic contexts, including the overexpression of Myc, the combination of Scribble depletion with activation of Ras in clones or the early clonal depletion of Scribble in eye disc. Flamingo depletion reduces proliferation rate and increases the rate of apoptosis in the winner clones, hence reducing their competitiveness up to forcing their full elimination (hence becoming now "loser"). This function of Flamingo in cell competition is specific of Flamingo as it cannot be recapitulated with other components of the PCP pathway, does not rely on interaction of Flamingo in trans, nor on the presence of its cadherin domain. Thus, this function is likely to rely on a non-canonical function of Flamingo which may rely on downstream GPCR signaling.This unexpected function of Flamingo is by itself very interesting. In the framework of cell competition, these results are also important as they describe, to my knowledge, one of the only genetic conditions that specifically affect the winner cells without any impact when depleted in the loser cells. Moreover, Flamingo do not just suppress the competitive advantage of winner clones, but even turn them in putative losers. This specificity, while not clearly understood at this stage, opens a lot of exciting mechanistic questions, but also a very interesting long term avenue for therapeutic purpose as targeting Flamingo should then affect very specifically the putative winner/oncogenic clones without any impact in WT cells.The data and the demonstration are very clean and compelling, with all the appropriate controls, proper quantifications and backed-up by observations in various tissues and genetic backgrounds. I don't see any weakness in the demonstration and all the points raised and claimed by the authors are all very well substantiated by the data. As such, I don't have any suggestions to reinforce the demonstration.While not necessary for the demonstration, documenting the subcellular localisation and levels of Flamingo in these different competition scenarios may have been relevant and provide some hints on a putative mechanism (specifically by comparing its localisation in winner and loser cells).

While we did not perform a thorough analysis, our current revision of the manuscript shows Fmi staining results that do not support a change in subcellular localization of Fmi. In our images, Fmi seemed to localize similarly along the winner-loser clone boundaries, and inside and outside the clones. We cannot rule out that a subtle change in localization is taking place that could perhaps be detected with higher resolution imaging.

Also, on a more interpretative note, the absence of impact of Flamingo depletion on JNK activation does not exclude some interesting genetic interactions. JNK output can be very contextual (for instance depending on Hippo pathway status), and it would be interesting in the future to check if Flamingo depletion could somehow alter the effect of JNK in the winner cells and promote downstream activation of apoptosis (which might normally be suppressed). It would be interesting to check if Flamingo depletion could have an impact in other contexts involving JNK activation or upon mild activation of JNK in clones.

See our comment to Reviewer 2 regarding JNK.

Strengths:A clean and compelling demonstration of the function of Flamingo in winner cells during cell competitionOne of the rare genetic conditions that affects very specifically winner cells without any impact in losers, and then can completely switch the outcome of competition (which opens an interesting therapeutic perspective on the long term) Weaknesses:The mechanistic understanding obviously remains quite limited at this stage especially since the signaling does not go through the PCP pathway.

We agree that in the future, it will be desirable to gain a mechanistic understanding of Fmi’s role in competition.

**Recommendations for the authors:**

**Reviewer #1 (Recommendations for the authors):**
I have read the revised manuscript and have found issues that need to be resolved. The biggest concern is the overstatement of the results that loss of fmi from Myc-overexpressing clones turns them into losers. This is not shown in a compelling manner in the revised manuscript and the authors need to tone down their language or perform more experiments to support their claims.(1) I do not agree with the language used by the authors last paragraph of p. 4 stating loss of fmi from Myc supercompetitors (Fig. 2) makes them losers. At this point in the paper, they only use clone size as a readout. By definition, losers in imaginal discs die by apoptosis, which is not measured in this figure. As such, the authors do not prove that fmi-mutant Myc over-expressing clones are now losers at this point in the manuscript. The authors should discuss this in the results section regarding Fig. 2.

We have modified the language in text and figure legend to acknowledge that the clone size data alone do not demonstrate competition.

(2) Related to point #1, I do not agree with the language in the legend of Fig. 2H that the graph is measuring "supercompetition". They are only measuring clone ratios, not apoptosis. Growing to a smaller size does not make a clone have loser status without also assessing cell death.(a) I suggest that the authors remove the sentence "A ratio over 0 indicates supercompetition of nGFP+ clones, and below 0 indicates nGFP+ cells are losers." in the legend to Fig. 2H. Instead, they should describe the assay in times of clone ratios.

The reviewer raises a valid point, as at this point in the manuscript we did not quantify cell death and proliferation. However, based on decades of knowledge of supercompetiton, Myc clones are classified as super-competitors in every instance they’ve been studied. (Myc clones show apoptosis when competing with WT cells, while at the same time they eliminate WT neighbors by apoptosis to become winners. Their faster proliferation rate may be what ultimately makes them winners.) We changed the language to address this distinction.

(3) In Fig. 4, they do attempt to monitor apoptosis, which is the fate of bona fide losers in imaginal tissue. However, I have several concerns about these data (panels 4I-K, O and P have been added to the revised manuscript.)(a) In Fig. 4I-K, why is there no death of WT cells which would be expected based on de la Cova Cell 2004? The authors need to comment on this.(b) Cell death should also be observed in the Myc over-expressing clones but none is seen in this disc (see de la Cova 2004 and PMID: 18257071 Fig. 4). The authors need to comment on this.

We do not understand why the reviewer raises these two points. We see some cell death in >Myc eye discs both in winners and losers, as displayed in the graph. In our hands, the levels were on average very low. The example shown is representative of the analysis and shows apoptosis both in WT and >Myc cells, highlighted by the arrows in 4J. We added a mention to the arrows in the figure legend to make it clearer. In the main text, we already compared our observations to the same publication the reviewer mentions (De la Cova 2004).

(c) The data in panel 4O is not explained sufficiently in the legend or results section. What do the lines between the data points in the left side of the panel mean? Why is there a bunch of clustered data points in the right part of the Fig. 4O, when two different genotypes are listed below? I would have expected two clusters of points. The authors need to comment on this.

We intended to convey as much information as possible in an informative manner in these graphs, and we regret not explaining better the analysis shown. We modified the legends for the apoptosis analysis to better explain the displayed data.

(d) What is the sample size (n) for the genotypes listed in this figure? The authors need to comment on this and explicitly list the sample size in the legend.

We added the n for both conditions to the figure.

(e) In panels 4L-N, why is the death occurring in the apparent center of the fmiE59>>Myc clone. If these clones are truly losers as the authors claim, then apoptosis should be seen at the boundaries between the fmiE59>>Myc clone and the WT clones. The results in this figure are not compelling, yet this is the critical piece of data to support their claim that fmiE59>>Myc clone are losers. The authors need to comment on this.

The majority of cell death in this example is observed 1-3 cells away from the clone boundary. In some cases, we observe cell death farther from the boundary, but those cells were not counted in our analyses. As described in our methods, we only considered for the analysis cells at the clone boundary or in the vicinity, as those are the ones that most probably have apoptosis triggered by the neighboring clone.

(f) There is no red line in Fig. 4O and 4P, in contrast to what is written in the legend in the revised manuscript. This should be corrected.

We thank the reviewer for catching the error about the line. We have now simplified the graph by removing the line at Y=0 and just leave one dashed line, representing the mean difference between WT and >>Myc cells.

(4) On p. 10, the reference Harvey and Tapon 2007 to support hpo-/- supercompetitor status is incorrect. The references are Ziosi 2010 and Neto-Silva 2010. This should be changed.

We thank the reviewer for the correction. While the review we provided discusses the role of the Hpo pathway in proliferation and cancer, it does not discuss competition. The reference we intended to include here was Ziosi 2010. We now cite both in the revised manuscript.

(5) The legend for Fig. 3A-H is missing from the revised manuscript. This needs to be added.

This was likely a copy-edit glitch. The missing parts of the legend have been restored.

(6) Material and methods is missing details on the hs-induced clones. The authors need to specifically state when the clones were generated and when they were analyzed in hours after egg laying.

The timing of the heat-shock and analysis was described in the methods: “Heat-shock was performed on late first instar and early second instar larvae, 48 hrs after egg laying (AEL). Vials were kept at 25ºC after heat-shock until larvae were dissected”. And additionally, in the dissection methods: “Third instar wandering larvae (120 hrs AEL) were dissected…” We have included in this revision the length of the heat-shock (15 min).

I have read the rebuttal and some of my concerns are not sufficiently addressed.(8) I raised the point of continuously-generated clones becoming large enough to evade competition, and I disagree with the authors' reply. I think that competition of RasV12, scrib (or lgl) competition largely depends the size of the clone, which is de facto larger when generated by continuous expression of flp (such as eyeless or tubulin promoters used in this study). I think that at that point, we are at an impasse with respect to this issue, but I wanted to register my disagreement for the record. Related to this, one possible reason for the fragmentation of the fmimutant Myc overexpressing clones in the wing disc is because they were not continuously generated and hence did not merge with other clones.

Please see the discussion above in the public comments. We remain unclear about what, exactly, the reviewer disagrees. As stated above, we think they are correct that the size of the clone is critical in determining winner vs loser status.

**Reviewer #2 (Recommendations for the authors):**
Although the authors have addressed some of my concerns, I still feel that a detailed mechanistic understanding is essential. I hope the authors will conduct additional experiments to solve this issue.

We also consider the mechanism of interest and will pursue this in the future. To test our hypotheses we require a set of genetic mutants that are still in the making that will help us dissect the function and potential partners of Fmi, and we hope to have these results in a future publication.

**Reviewer #3 (Recommendations for the authors):**
- There is no clear demonstration that the relative decrease of clone size in UASMyc/Fmi mutant is mostly driven by either a context dependant suppression of growth and/or an increase of apoptosis (the latter being the more classic feature of loser phenotype).

We believe that it is driven by both, and refrain from making assumptions about the magnitude of contribution from each. This question is something that we will be interested to explore in the future.

The distribution of cell death in Fmi/UAS-Myc mutant is somehow surprising and may not fit with most of the competition scenarios where death is mostly restricted to clone periphery (although this may be quite variable and would require much more quantification to be clear).

While we observe some cell death far from clone boundaries, most of the dying cells are a few cells away from a clone boundary. In other publications quantifying cell death, examples of cell death farther from the boundary are not rare (See for example Moreno and Basler 2004 Fig 6, De la Cova et al. Fig 2, Meyer et al 2014 Fig 2). We did not count cells dying far from clone boundaries in our analysis.

I just noticed a few mistakes in the legend :Figure 3M legend is missing (it would be useful to know at which stage the quantification is performed)

Another reviewer brought to our attention the problems with Fig 3 legend. We restored the missing parts.

It would be good to give an estimate of the number of larvae observed when showing the representative cases in Figure 1 .

This is a good point. We now include these numbers in the figure legend.